# CO₂-dependent opening of connexin 43 hemichannels

**Valentin Mihai Dospinescu[†], Alexander Mascarenhas[†], Jack Butler, Sarbjit Nijjar, Kyara de Oliveira Taborda, Sean Connors, Lumei Huang, Nicholas Dale***

School of Life Sciences, University of Warwick, Coventry, United Kingdom

**eLife Assessment**

This **important** study reveals that connexin43 (Cx43) hemichannels are directly activated by CO₂ through a conserved carbamylation motif, extending a mechanism previously described for β-connexins to α-connexins. The evidence is **convincing**, supported by complementary biochemical and electrophysiological analyses showing CO₂-induced hemichannel opening and ATP release in cultured cells and hippocampal slices. These findings advance our understanding of connexin regulation by metabolic gases and will be of broad interest to researchers studying cell communication, neural signaling, and gasotransmitter biology.

**\*For correspondence:**
n.e.dale@warwick.ac.uk

[†]These authors contributed equally to this work

**Competing interest:** The authors declare that no competing interests exist.

**Abstract** Sequence and structure comparisons between alpha and beta connexins, Cx43 and Cx26, revealed that Cx43 has a motif, the carbamylation motif, that confers CO₂-sensitivity on a subset of beta connexins. By using a fluorescent dye loading assay, whole cell patch clamp recordings and real-time measurement of ATP release via GRAB$_{ATP}$, we have demonstrated that Cx43 hemichannels open in a highly CO₂-sensitive manner over the range 20–70 mmHg. Mutational analysis confirms that the equivalent residues to those in Cx26, known to be involved in mediating the effects of CO₂ on gating of hemichannels and gap junction channels, also mediate Cx43 hemichannel gating. These data predict that Cx43 will be partially open at resting physiological levels of PCO₂. In acute mouse hippocampal slices, we have demonstrated a CO₂-dependent enhancement of synaptic transmission that was blocked by the Cx43-selective mimetic peptide Gap26. Our data resolves an inconsistency in the literature between in vivo studies suggesting that Cx43 hemichannels are at least partially open at rest and in vitro studies performed in the absence of HCO₃⁻/CO₂ buffering that show Cx43 hemichannels are shut.

## Introduction

There are 21 connexin genes encoded in the human genome. Connexins are transmembrane proteins that can form hemichannels and gap junction channels. The existence of 21 isoforms suggests functional specialisation across the gene family (*Söhl and Willecke, 2004*). On the basis of molecular phylogeny, the connexins have been divided into four broad clades - the alpha, beta, gamma, and delta groupings (*Mikalsen et al., 2021*).

Certain members of the β-clade, Cx26, Cx30, and Cx32, are directly sensitive to CO₂ (*Huckstepp et al., 2010*; *Dospinescu et al., 2019*). The mechanism of CO₂ sensitivity has best been studied in Cx26. While Cx26 hemichannels are opened in response to CO₂ (*Huckstepp et al., 2010*), CO₂ closes Cx26 gap junction channels (*Nijjar et al., 2021*). The CO₂-dependent gating of Cx26 hemichannels and gap junction channels involves carbamylation of a specific lysine, K125, which is part of a critical carbamylation motif, $_{125}$KVRIEGS$_{131}$ (*Meigh et al., 2013*). This motif is essential for the CO₂ sensitivity of Cx26, as it allows the formation of a salt bridge between the carbamylated K125 and R104 of the

neighbouring subunit (*Meigh et al., 2013*). The formation of this 'carbamate' bridge induces conformational changes, ultimately leading to the opening of the hemichannel or closing of the gap junction channel (*Meigh et al., 2013*; *Nijjar et al., 2021*; *Brotherton et al., 2022*; *Brotherton et al., 2024*). Mutation of K125 to arginine, an amino acid that cannot be carbamylated, or R104 to alanine so that it cannot form a carbamate bridge, removes the ability of $CO_2$ to open Cx26 hemichannels (*Meigh et al., 2013*) or close Cx26 gap junction channels (*Nijjar et al., 2021*; *Nijjar et al., 2025*). Furthermore, the introduction of the carbamylation motif into a $CO_2$-insensitive connexin, Cx31, results in $CO_2$-dependent hemichannel opening of Cx31 (*Meigh et al., 2013*). Other members of the β-clade, Cx30, and Cx32, are also $CO_2$-sensitive, but have different sensitivity profiles (*Huckstepp et al., 2010*; *Dospinescu et al., 2019*). Evolutionary comparisons show that the carbamylation motif has been conserved over 400 million years (*Dospinescu et al., 2019*) and appears to be a necessary and sufficient feature for the $CO_2$-sensitivity of connexins.

Sequence comparisons of the β-connexins and α-connexins revealed the presence of a potential carbamylation motif within the α-connexin, Cx43. Cx43 has a motif ($_{144}$KVKMRGG$_{150}$) homologous to the motif in Cx26 ($_{125}$KVRIEGS$_{131}$) suggesting that it might also be sensitive to $CO_2$. In this study, we show that Cx43 hemichannels are indeed directly sensitive to changes in the partial pressure of $CO_2$ ($PCO_2$) over the physiological concentration range. Unlike Cx26, the $CO_2$-dependent opening mechanism of Cx43 is more complex, involving additional residues beyond the homologs of K125 and R104 in Cx26. Our discovery that Cx43 hemichannels are directly $CO_2$ sensitive has major physiological implications and may explain why Cx43 hemichannels appear to be open under physiological conditions but closed in vitro where they are usually studied in the absence of $CO_2/HCO_3^-$ buffers.

## Results

### Identification of a potential carbamylation motif in Cx43

Cx43, an alpha-connexin, possesses a carbamylation motif homologous to that of the beta-connexins (*Figure 1A*). AlphaFold3 (AF3) predicts a structure very close to the experimentally determined structures for Cx43 (*Lee et al., 2023a*; *Qi et al., 2023*) but has the advantage of including portions of the structure that have not been experimentally resolved. This is a valid approach, as the de novo predictions by AF3 for the cytoplasmic loop of Cx26 have been supported by a recent experimentally derived structure (*Brotherton et al., 2024*). By performing alignments of the Cx43 and Cx26 AF3 structures (*Figure 1B*), we found that two residues in Cx43, K144, and K105, aligned with the residues in Cx26 that are essential for hemichannel opening to $CO_2$, K125, and R104 (*Figure 1C*). The predicted 3D orientation of the alpha-helices involved are positioned in a similar manner to allow potential interaction between K144 and K105 following carbamylation. Cx43 also presents another feature that is similar to Cx26: the high incidence of Lys residues. In Cx26, five lysine residues in the cytoplasmic loop can be carbamylated (*Nijjar et al., 2025*), suggesting an environment favourable for this to happen. The presence of multiple Lys residues in the cytoplasmic loop of Cx43 implies that the local environment may be conserved to facilitate the required pKa for $CO_2$ binding and carbamate bridge formation.

### Cx43 hemichannels are opened by $CO_2$

We used an established dye-loading protocol, previously used for $CO_2$ connexin studies (*Huckstepp et al., 2010*; *Meigh et al., 2013*; *Meigh et al., 2014*; *de Wolf et al., 2016*; *de Wolf et al., 2017*; *Cook et al., 2019*; *Dospinescu et al., 2019*; *van de Wiel et al., 2020*), to test whether Cx43 hemichannels are $CO_2$ sensitive. We observed significant dye-loading in Cx43-expressing cells when exposed to a high $CO_2$ stimulus (70 mmHg) when starting from a baseline of 20 mmHg (MW test, $PCO_2$ 20 vs 70 mmHg: p=0.008, n=5, *Figure 2A*). We used a zero $Ca^{2+}$ aCSF to unblock hemichannels as a positive control to demonstrate that for WT Cx43, the zero $Ca^{2+}$ stimulus and 70 mmHg $PCO_2$ gave a similar amount of dye loading (MW test: p-value = 0.726, n=5).

To further support these findings, we next used whole cell patch clamp recordings from HeLa cells expressing Cx43 (*Figure 2B*). We set the holding potential to –50 mV and gave regular steps to –40 mV to measure whole cell conductance. From a baseline of 35 mmHg $PCO_2$, an increase in outward conductance was observed when switching to 55–70 mmHg aCSF (*Figure 2B*). On switching to 20 mmHg aCSF, a small conductance decrease and reduction of outward current was observed

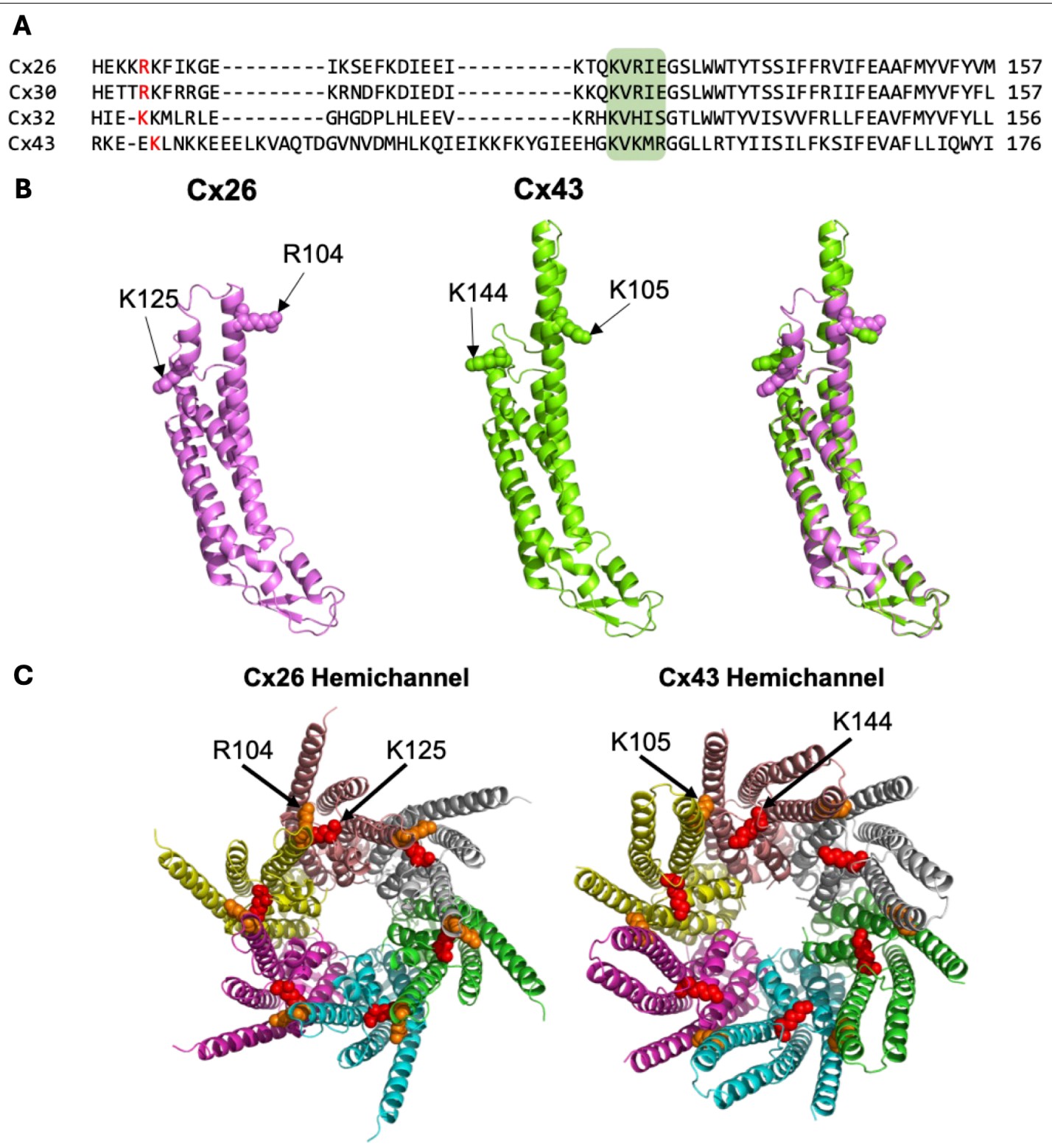

**Figure 1.** Structures of Cx43 and Cx26 predicted by AF3. (**A**) α- and β-connexin sequence alignment with the known $CO_2$-sensitive connexins at the top, Cx26, Cx30, and Cx32, followed by the α-connexin Cx43. The carbamylation motif is highlighted by the green box. (**B**) Cx26 and Cx43 AlphaFold3 structural alignments. (**C**) Cx26 and Cx43 hemichannel structural prediction.

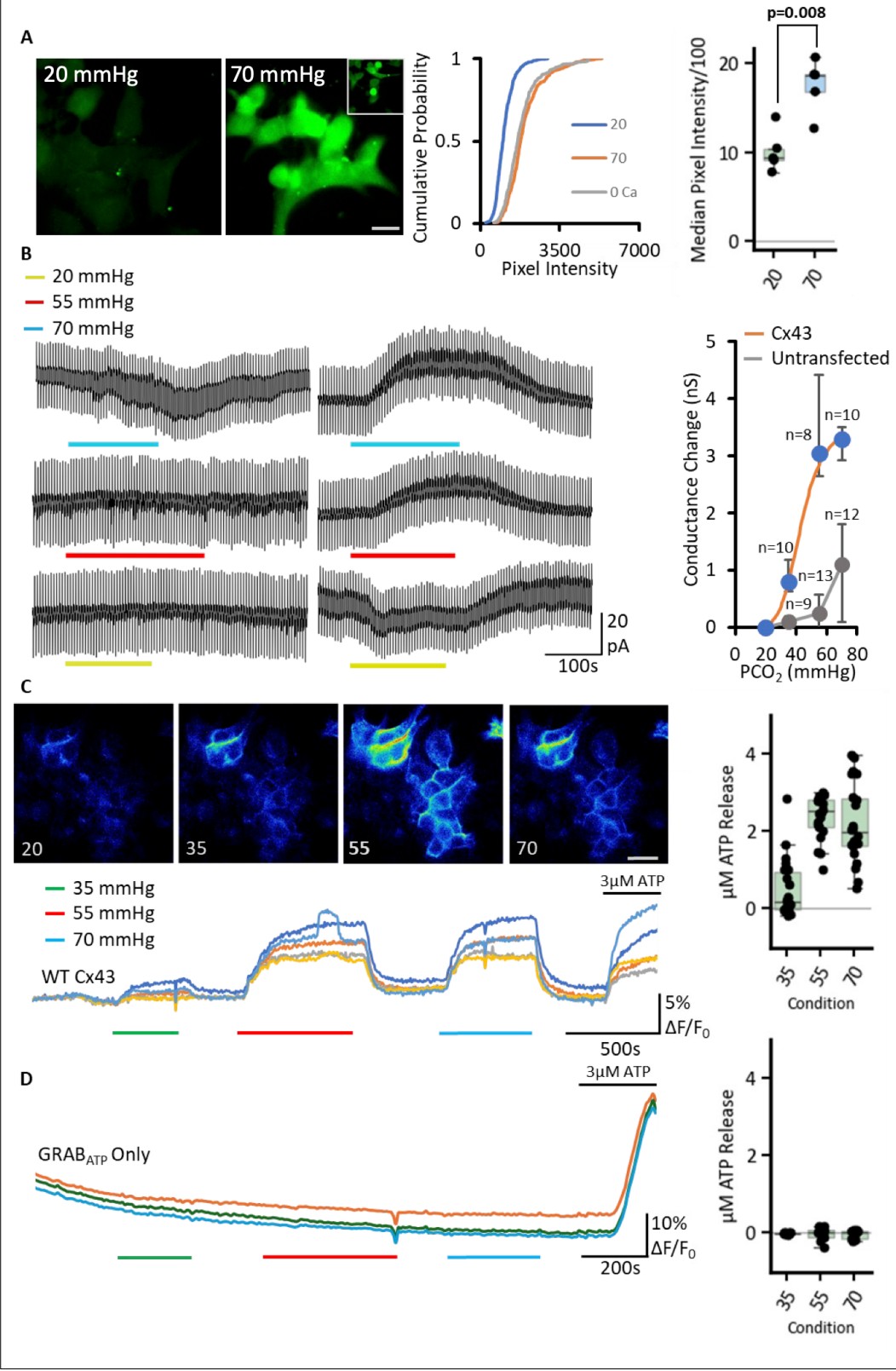

**Figure 2.** Cx43 hemichannels can be opened by an increase in PCO$_2$. The CO$_2$ sensitivity of Cx43 was assayed using three different methods. (**A**) Dye-loading with CBF from 20 (left) to 70 mmHg PCO$_2$ (right). The inset represents the 0 Ca$^{2+}$ control, white scale bar is 20 μm. Cumulative probability graph represents all the data points measured with more than 200 cells per condition, box plot shows the medians from each independent

*Figure 2 continued on next page*

*Figure 2 continued*

transfection. (**B**) Whole cell patch recordings performed on untransfected parental HeLa cells and HeLa cells expressing Cx43 at a holding potential of –50 mV with steps to –40 mV to measure whole cell conductance. The control level of $PCO_2$ was 35 mmHg and switched to the three different concentrations were used (20, 55, 70 mmHg, indicated by colour bars). The absolute conductance change evoked by a change in $PCO_2$ was measured and plotted as a dose response curve (right side, median with interquartile range) with 20 mmHg being assigned to zero and the absolute value of all conductance changes plotted relative to this (see Methods for details). The points were fitted by the Hill equation H=6, $EC_{50}$=43 mmHg (blue points, orange curve). The data for untransfected parental HeLa cells was plotted the same way (grey). (**C**) The genetically encoded ATP sensor $GRAB_{ATP}$ was co-transfected alongside Cx43 into HeLa cells (n=22). Representative images at different $PCO_2$ concentrations of the cells are shown, grey scale bar is 20 µm. Traces from the recording the images were selected from are shown; each line represents a different cell, the fluorescence was normalised by dividing all values by the baseline median pixel intensity before plotting. Box plot shows the µM ATP release as determined by normalisation to the 3 µM control solution applied. (**D**) Parental HeLa cells transfected only with $GRAB_{ATP}$ do not exhibit $CO_2$-dependent ATP release (n=14), summary box plot shows the µM ATP release as determined by normalisation to 3 µM ATP calibration.

The online version of this article includes the following source data and figure supplement(s) for figure 2:

**Source data 1.** All the source data for the graphs in *Figure 2* and its supplements.

**Figure supplement 1.** Cx43 hemichannels with a truncated C-terminus (Cx43$^{1-256}$) retain their sensitivity to $CO_2$ and depolarisation.

**Figure supplement 2.** Effect of hypercapnic solutions on intracellular pH ($pH_i$) of parental HeLa cells measured by BCECF fluorescence.

**Figure supplement 3.** Cx43 gap junction channels are insensitive to changes in $PCO_2$.

(*Figure 2B*). Untransfected parental HeLa cells did not display changes in outward current to changes in $PCO_2$. However, some parental HeLa cells exhibited a slow increase in inward current on transfer from 35 to 70 mmHg (*Figure 2B*). This was quite distinct from the outward current observed in Cx43 expressing cells (*Figure 2B*). Overall, the patch clamp data predicted a dose response that could be fitted by the Hill equation with a Hill coefficient of 6 and an $EC_{50}$ of 43 mmHg. This is similar to Cx26 hemichannels which also display a steep $CO_2$ dose response curve with a Hill coefficient of >4 (*Huckstepp et al., 2010*; *de Wolf et al., 2017*; *van de Wiel et al., 2020*).

To assess whether the $CO_2$-dependent opening of Cx43 resulted in ATP release, we co-transfected HeLa cells with the genetically encoded ATP sensor - $GRAB_{ATP}$ (*Wu et al., 2022*). A $CO_2$-dependent increase in ATP release was indeed observed (n=22, *Figure 2C*). We observed an increase in $GRAB_{ATP}$ fluorescence on going from 20 to 35 mmHg $PCO_2$ and further to 55 mmHg $PCO_2$ (*Figure 2C*). While ATP release at 70 mmHg was slightly less than that at 55 mmHg, it remained markedly higher than at 35 mmHg $PCO_2$ (*Figure 2C*). By contrast, HeLa cells that were not transfected with Cx43 showed no responses to any applied $CO_2$ concentration as concluded from $GRAB_{ATP}$ experiments (n=14, *Figure 2D*). Overall, our results show through three independent assays that Cx43 hemichannels are opened by modest changes in $PCO_2$ around the physiological norm and will be partially open at a $PCO_2$ of 35 mmHg (*Figure 2*). Once opened by $CO_2$, Cx43 hemichannels display permeability to ATP (*Figure 2C*).

Cx43 displays sensitivity to intracellular pH via a sensor located in the C-terminus (*Ek-Vitorín et al., 1996*). We assessed the effect of truncations of the C-terminus of Cx43 (truncated after residue 256, Cx43$^{1-256}$) that remove this pH sensor. Both $CO_2$ and high K$^+$induced ATP release from HeLa cells that expressed Cx43$^{1-256}$ (*Figure 2—figure supplement 1*). We also measured the effect of changing $PCO_2$ on intracellular pH ($pH_i$) of HeLa cells. We found that transferring from 35 to 55 or 70 mmHg, respectively, induced median changes in $pH_i$ of –0.02 (95% CI –0.02,–0.03) and –0.13 (CI –0.10,–0.15; *Figure 2—figure supplement 2*). These results show that the effects of $PCO_2$ on Cx43 gating are independent of those involved in its pH sensitivity.

Gap Junction Channels (GJCs) of Cx26 are closed by $CO_2$ also acting via the carbamylation motif. We therefore tested whether Cx43 GJCs might also be $CO_2$ sensitive by using a dye transfer assay (*Nijjar et al., 2021*). Cx43 GJCs allowed equally fast permeation of dye at all levels of $PCO_2$ and therefore, unlike the hemichannel, were insensitive to $CO_2$ (*Figure 2—figure supplement 3*).

## Identifying the residues required for $CO_2$-sensitivity of Cx43

Having established the dose-response characteristics of Cx43 to $CO_2$, we next examined the potential underlying mechanisms for $CO_2$-dependent gating via targeted mutagenesis of possible key residues, starting with the equivalents of those known to be involved in the $CO_2$ sensitivity of Cx26 (*Figure 1*).

Mutation of K144 to glutamine (K144Q), an amino acid that cannot be carbamylated, partially reduced the $CO_2$ sensitivity of the channel: the 70 mmHg stimulus resulted in significantly less increase in dye loading than the zero $Ca^{2+}$-positive control (*Figure 3A*, MW test: p=0.048, n=5). However, mutation of Lys105 to a non-charged residue (K105Q) resulted in hemichannels that gave a similar increase in dye loading to the zero $Ca^{2+}$-positive control (*Figure 3B*, MW test: p=0.111, n=5). In Cx26, mutation of the residues involved in carbamylation (K125 and R104) results in complete loss of $CO_2$ sensitivity (*Meigh et al., 2013*; *Nijjar et al., 2021*). The lack of such a complete effect on the $CO_2$ sensitivity of Cx43 implies that the mechanism is more complex and may involve further residues. Interestingly, mutation of Lys144 to Arg (K144R) completely abolished the $CO_2$ sensitivity of Cx43 hemichannels as measured by dye loading (*Figure 3—figure supplement 1*). The difference between the effect of mutating Lys144 to Arg and Gln on $CO_2$ sensitivity may arise because the substitution to Arg introduces a positive charge. Since K144 appears to be carbamylated, its primary amine must remain unprotonated so that the unpaired electrons of the nitrogen can mount a nucleophilic attack on the carbon of $CO_2$ to form the carbamate bond. Thus, the K144Q mutation may preserve this charge and be a better substitute that removes the capacity for carbamylation while preserving the overall charge within this region of the protein. Consequently, we have predominantly used the Lys to Gln substitution in this study.

We further inspected the AF3 structure prediction and identified two further lysine residues of interest (*Figure 3E*). K109 in Cx43 aligns with K108 of Cx26, a residue known to be carbamylated and required for Cx26 gap junction closure (*Nijjar et al., 2025*). AF3 also suggests that K109 could potentially interact with K234 following any carbamylation (*Figure 3E*). We therefore mutated both residues to Gln to test their possible involvement in $CO_2$ sensitivity. The dye-loading assay showed that K109Q remained $CO_2$ sensitive (*Figure 3C* MW test vs zero $Ca^{2+}$ control, p=0.274, n=5). However, the mutation K234Q displayed impaired $CO_2$ sensitivity (*Figure 3D*, MW test vs zero $Ca^{2+}$ control: p=0.016, n=5).

As the dye loading assay suggested that single mutations were on the whole rather ineffective in abolishing $CO_2$ sensitivity, we assayed ATP release via $GRAB_{ATP}$ to gain further supporting evidence. We found that $CO_2$-evoked ATP release remained dose dependent and very similar to the WT Cx43 in the K105Q (n=23) and K144Q (n=12) mutants (*Figure 4A, C*, *Figure 4—figure supplement 1*). Cx43 hemichannels with the K109Q mutation appeared to have their sensitivity to $CO_2$ shifted to lower values of $PCO_2$ (n=20), appearing to give maximal ATP release at 35 mmHg (*Figure 4B, E*, *Figure 4—figure supplement 1*). K234Q Cx43 hemichannels had greatly reduced sensitivity to $CO_2$ (n=15, *Figure 4D and F*). To check that K234Q did not have some non-specific effect on hemichannel function, we used high $K^+$-induced depolarisation as a way of opening the hemichannel independently of changes in $PCO_2$ as a positive control. This stimulus resulted in robust ATP release.

## $CO_2$ sensitivity of Cx43 involves multiple Lys residues

As single Lys mutations mostly had partial effects on the $CO_2$ sensitivity of Cx43 hemichannels, we next mutated pairs of Lys residues. Using the dye loading assay, we found that all combinations of paired mutations abolished $CO_2$ sensitivity relative to the zero $Ca^{2+}$ control (*Figure 5*): K109Q, K144Q (MW test p=0.008, n=5); K105Q, K144Q (MW test p=0.004, n=5); K105Q, K109Q (MW test p=0.004, n=5); K105Q, K234Q (MW test p=0.004, n=5); and K144Q, K234Q (MW test p=0.004, n=5). Using the $GRAB_{ATP}$ assay, we confirmed that all these double mutations also abolished $CO_2$-dependent ATP release (MW test for all mutations, $CO_2$ vs high $K^+$-positive control $p<10^{-6}$, *Figure 6*).

## The effect of introducing negatively charged residues into the carbamylation motif

We next sought to mimic carbamylation by mutating Lys to Glu. This introduces into the motif the negative charge that would occur upon carbamylation of Lys. In the dye loading assay, the mutation K144E resulted in a complete loss of $CO_2$ sensitivity while dye loading to zero $Ca^{2+}$ was retained (MW test, 70 mmHg vs zero $Ca^{2+}$: p=0.008, n=5, *Figure 7B*). The mutation K105E seemed to disrupt both

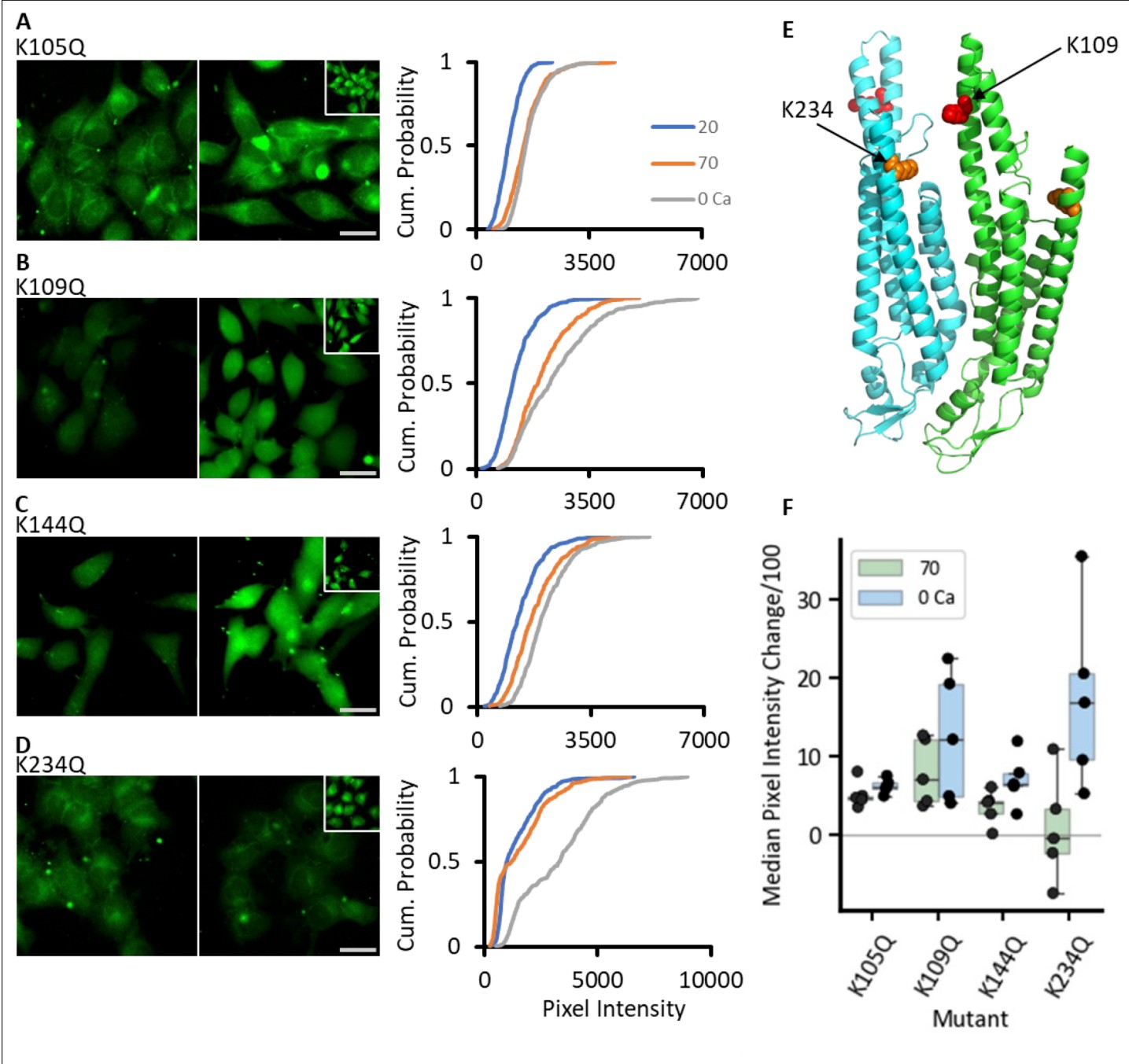

**Figure 3.** The effect of single Lys mutations on the $CO_2$ sensitivity of Cx43. HeLa cells transfected with the mutant variant of Cx43 were subjected to the dye-loading protocol. Images of dye loading in response to $CO_2$ and the 0 $Ca^{2+}$-positive control (insets), together with the cumulative probability plots are shown for K105Q (**A**), K109Q (**B**), K144Q (**C**), K234Q (**D**). (**E**) AlphaFold3 prediction of Cx43 with hypothesised carbamylation bridge residues highlighted – K234 in orange and K109 in red. (**F**) Summary box plots for dye-loading showing the change in median pixel intensity from 20 mmHg $PCO_2$ for each condition (70 mmHg $PCO_2$ – green and 0 $Ca^{2+}$ blue, n=5) for WT (recalculated data from **Figure 2A** for comparison) and each mutation. Scale bars are 20 µm.

The online version of this article includes the following source data and figure supplement(s) for figure 3:

**Source data 1.** All the source data for the graphs in **Figure 3** and its supplements.

**Figure supplement 1.** Cx43^K144R hemichannels are insensitive to changes in $PCO_2$.

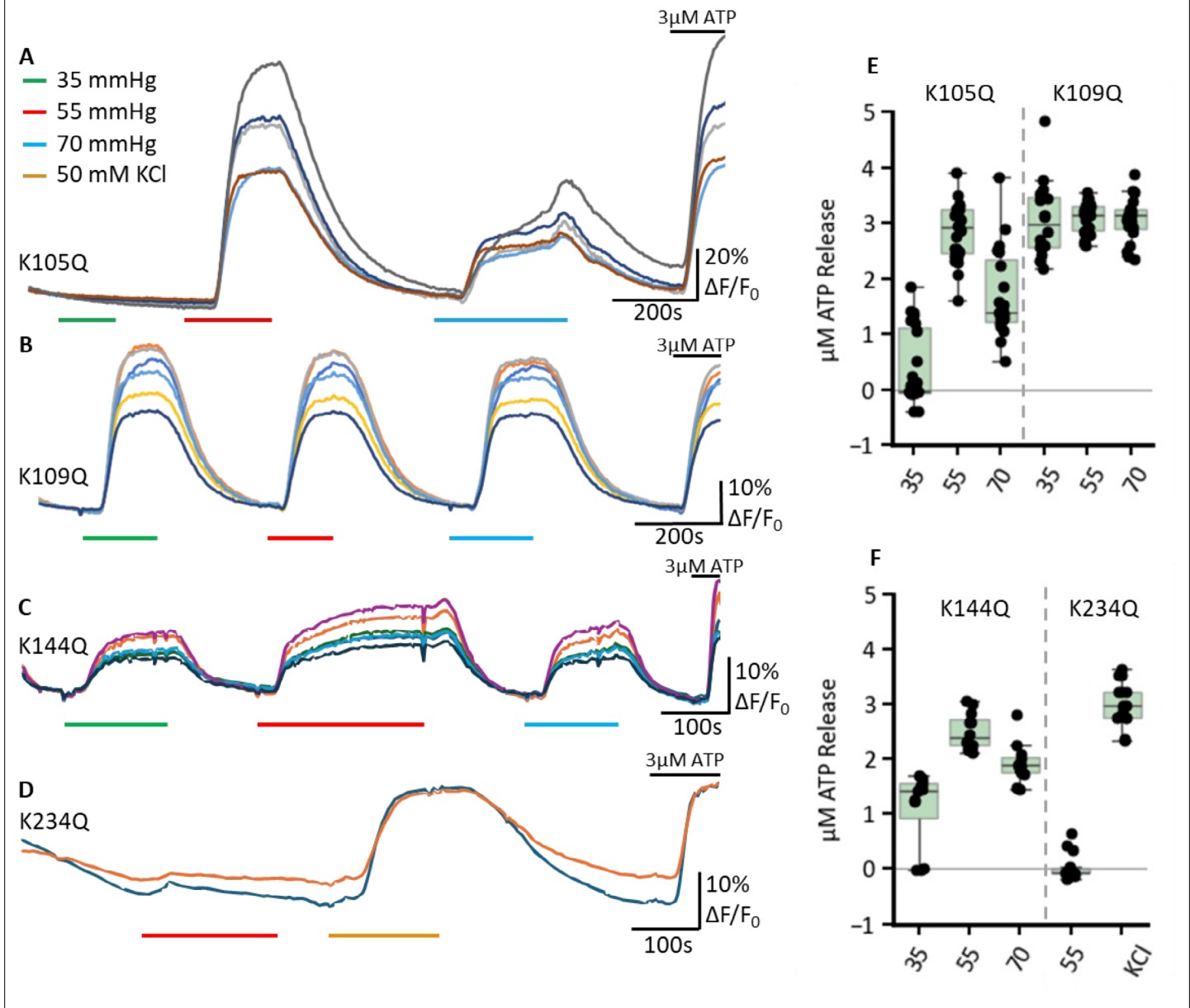

**Figure 4.** The effect of single Lys mutations on $CO_2$-dependent ATP release from Cx43. The genetically encoded ATP sensor $GRAB_{ATP}$ was co-transfected alongside Cx43 mutant variations, K105Q (**A**) n=23, K109Q (**B**) n=20, K144Q (**C**) n=12, K234Q (**D**) n=15. Representative trace measurements were selected; each line represents a different cell; the control level of $PCO_2$ was 20 mmHg and switched to higher values indicated by coloured bars. The fluorescence was normalised by dividing all values by the baseline median pixel intensity before plotting. Box plot on the right (**E**) and (**F**) represent the µM ATP released as determined through normalisation to the 3 µM ATP application.

The online version of this article includes the following source data and figure supplement(s) for figure 4:

**Source data 1.** All the source data for the graphs in *Figure 4*.

**Figure supplement 1.** The effect of single Lys mutations on the $CO_2$ dose-response properties of Cx43.

**Figure supplement 1—source data 1.** Parameters for the curves in *Figure 4—figure supplement 1*.

the $CO_2$-dependent and the zero $Ca^{2+}$ evoked dye loading (*Figure 7A*). K234E displayed impaired $CO_2$-dependent dye loading compared to that evoked by zero $Ca^{2+}$ (MW test, 70 mmHg vs zero $Ca^{2+}$: p=0.048, n=5, *Figure 7C*).

Using the $GRAB_{ATP}$ assay, we found that each of these single Lys to Glu mutations disrupted $CO_2$-dependent ATP release, but did not prevent depolarisation evoked ATP release (*Figure 8*).

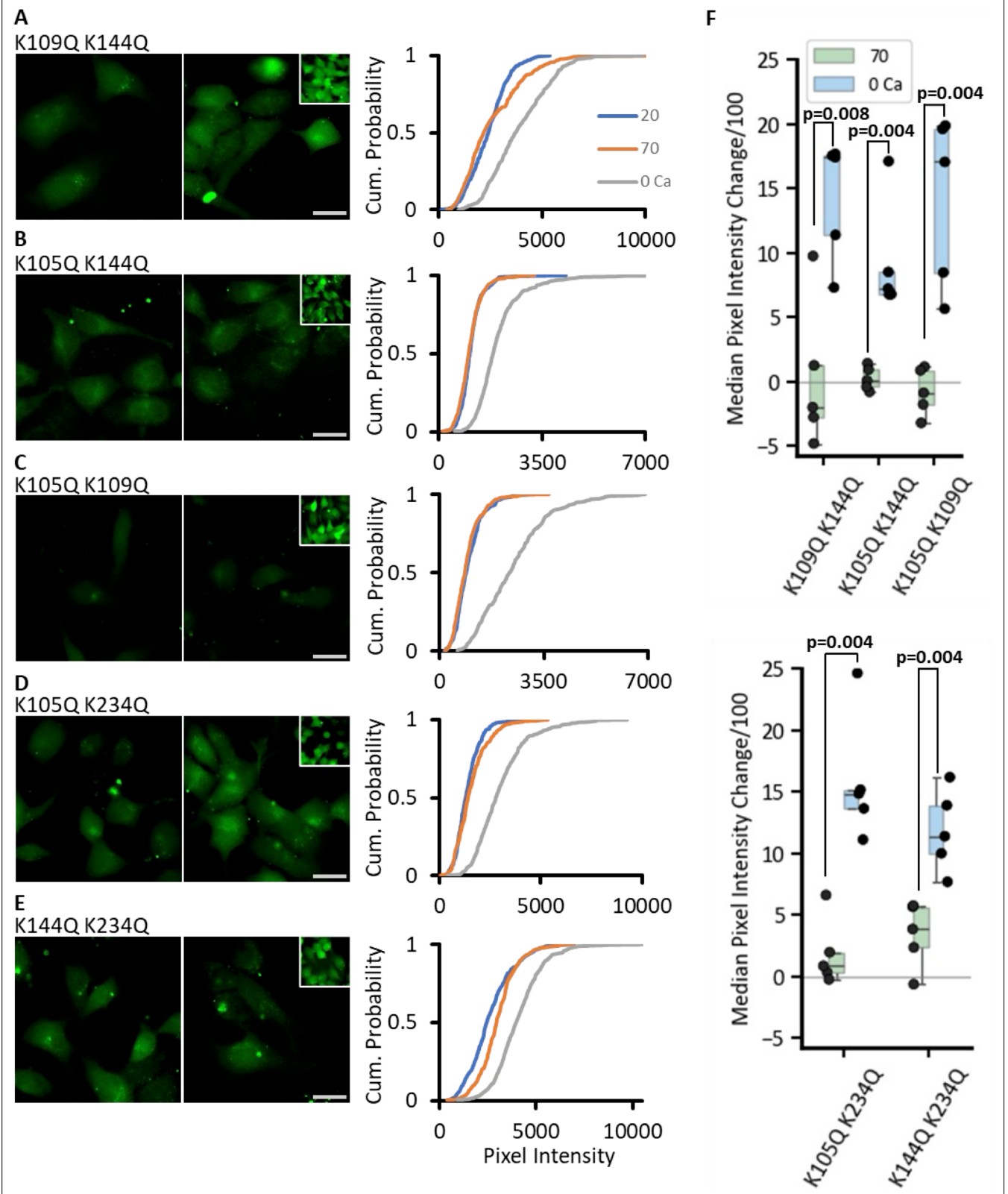

**Figure 5.** Paired Lys mutations are required to abolish the $CO_2$ sensitivity of Cx43. Representative images for all combinations tried are shown – K109Q K144Q (**A**), K105Q K144Q (**B**), K105Q K109Q (**C**), K105Q K234Q (**D**), K144Q K234Q (**E**). Insets represent the 0 $Ca^{2+}$-positive control. For each construct, the pixel intensity in expressing cells was measured from five individual transfections, with at least 40 cells per condition. Cumulative probability

*Figure 5 continued on next page*

*Figure 5 continued*

plots display all measured data points. Scale bar is 20 µm. (**F**) The box plots show the change in median pixel intensity from 20 mmHg $PCO_2$ for each transfection for 70 mmHg $PCO_2$ (green) and 0 $Ca^{2+}$ (blue).

The online version of this article includes the following source data for figure 5:

**Source data 1.** All the source data for the graphs in *Figure 5*.

This disruptive effect on $CO_2$ dependence might arise because introduction of the negative charge changes the local environment and makes it harder to carbamylate the other Lys residues.

As a change in $PCO_2$ would likely carbamylate more than one Lys residue simultaneously, we made the double mutation K105E and K109E. With the dye loading assay, we found that even at a $PCO_2$ of 20 mmHg, the cells loaded with dye suggesting that the mutated hemichannels were constitutively open (*Figure 9A and D*). Despite this, cells that expressed this double mutant had normal morphology (*Figure 9—video 1*) and their cultures did not exhibit noticeably higher levels of cell death. Presumably, the HeLa cells were able to adapt to the presence of these open channels.

Increasing $PCO_2$ to 70 mmHg gave no further dye loading. To test whether the channels were open even at a $PCO_2$ of 20 mmHg, we used the general hemichannel blocker $La^{3+}$ (100 µM) to show that this reduced the dye loading at 20 mmHg (*Figure 9B and D*, MW test: p=0.016, n=5). We used the GRAB$_{ATP}$ assay to demonstrate that at a $PCO_2$ of 20 mmHg, the K105E, K109E mutation caused continual release of ATP that could be blocked by $La^{3+}$ (*Figure 9C and D*, MW test 20 mmHg vs 20 mmHg + $La^{3+}$, p=$7.7 \times 10^{-7}$, n=16; *Figure 9—video 1*). This continual release of ATP at a $PCO_2$ of 20 mmHg was not evident from cells expressing WT Cx43, as application of 100 µM $La^{3+}$ had no effect on the GRAB$_{ATP}$ fluorescence (*Figure 9C and D*).

## Physiological role for $CO_2$-dependent ATP release via Cx43 from astrocytes

Cx43 hemichannels are $CO_2$ sensitive and are likely to be partially open at typical levels of $PCO_2$ in mammalian tissue. They could thus be able to release ATP under physiological resting conditions (35 mmHg $PCO_2$, *Figure 2*). Considering the ubiquitous distribution of Cx43 in astrocytes (*de Ceglia et al., 2023*) and the likely resting $PCO_2$ in the brain (*Hogg et al., 1984*), we postulated that, barring external modulatory influences, Cx43 hemichannels could continually release low levels of ATP in the hippocampus. Indeed, there exists compelling evidence for astrocytic Cx43 hemichannels being partially open under resting physiological conditions in vitro in the hippocampus (*Chever et al., 2014*). In this study, when Cx43 hemichannels were blocked by the mimetic peptide, Gap26, there was a decrease in synaptic strength of about 30% that was occluded by targeted knock out of astrocytic Cx43 expression (*Chever et al., 2014*).

As many investigators state that Cx43 hemichannels are shut under physiological conditions, we tested whether the results of *Chever et al., 2014* could be explained by the $CO_2$ sensitivity of Cx43. We predicted that if $PCO_2$ were lowered sufficiently to largely close Cx43, when $PCO_2$ was returned to the physiological norm, Cx43 should re-open and result in an increase of synaptic strength, presumably downstream of Cx43-dependent ATP release (*Chever et al., 2014*). Because ATP can be rapidly broken down to adenosine in the extracellular space (*Frenguelli et al., 2007*; *Wall and Dale, 2013*), which could then act at A1 receptors to inhibit transmission and give confounding effects, we used 8-CPT (8-cyclopentyltheophylline) to selectively block A1 receptors.

We preincubated hippocampal slices in 20 mmHg $PCO_2$ aCSF for 30 min prior to recording field EPSPs (fEPSPs) in area CA1 evoked by stimulation of the Schaffer collaterals. At constant extracellular pH, in the presence of 8-CPT, elevation of $PCO_2$ from 20 to 35 mmHg resulted in a 26% increase in the magnitude of the fEPSP (n=6, *Figure 10A*). To test whether this effect was mediated via $CO_2$-dependent opening of Cx43, we used 100 µM Gap26 to see whether this blocked the increase in fEPSP induced by 35 mmHg $PCO_2$. Once again, the fEPSP increased by about 30% on transfer to 35 mmHg aCSF (*Figure 10B*), and application of Gap26 reduced the fEPSP to below its baseline value (*Figure 10B and D*, Friedmann two-way ANOVA: p=0.006, n=6). This was an effect specific to the Gap26 peptide as 100 µM of the scrambled peptide did not have significant effects on the amplitude of the fEPSP (*Figure 10C and D*, Friedmann two-way ANOVA: p=0.55, n=5).

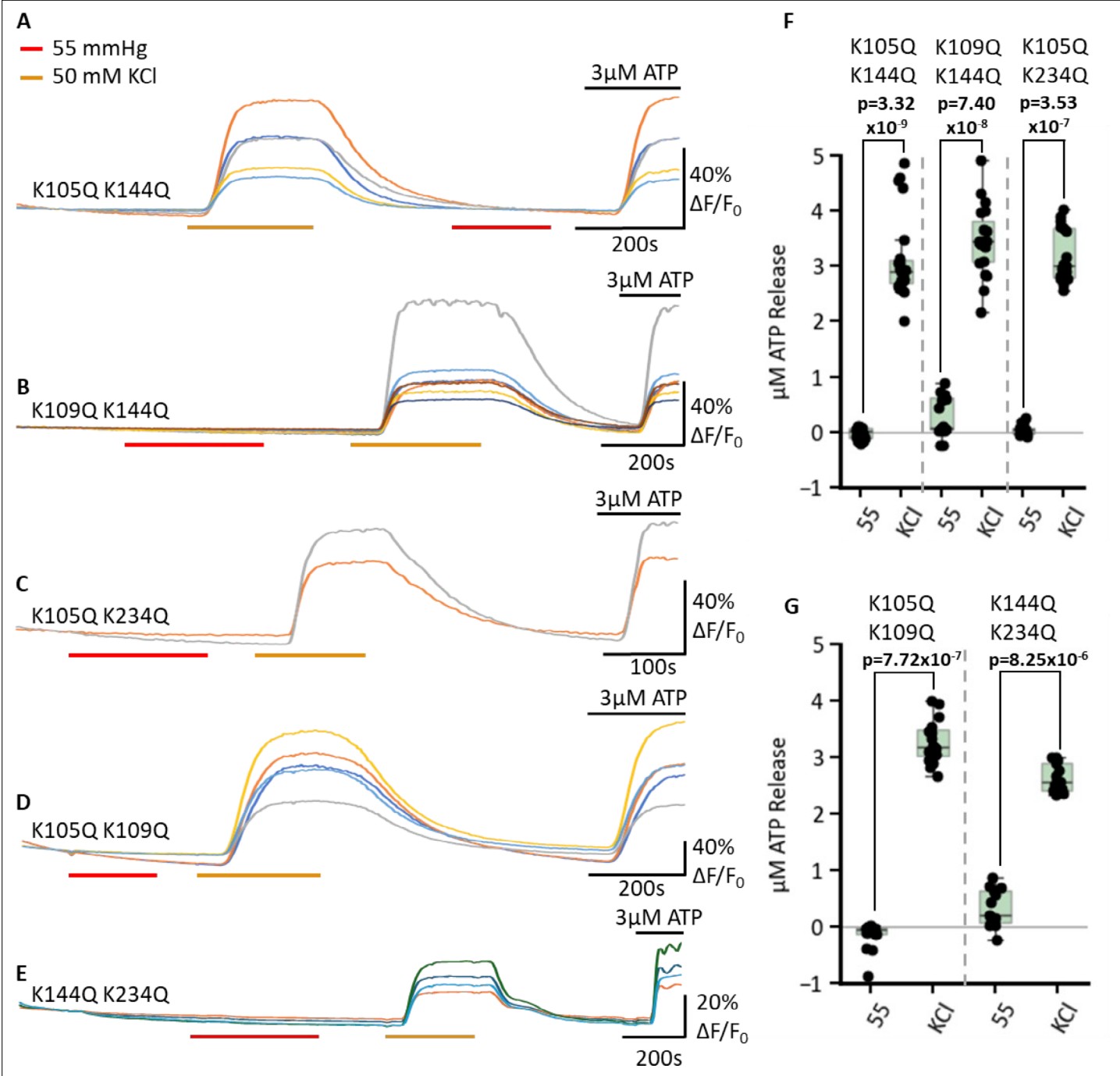

**Figure 6.** Paired Lys mutations abolish $CO_2$ dependent ATP release via Cx43. HeLa cells co-expressing Cx43 and $GRAB_{ATP}$ were subjected to changing $PCO_2$-levels and fluorescence was recorded. Representative traces (**A–E**) for each of the Cx43 mutations – K105Q K144Q (**A**) n=23, K109Q K144Q (**B**) n=19, K105Q K234Q (**C**) n=17, K109Q K234Q (**D**) n=16, K144Q K234Q (**E**) n=13. The traces indicate normalised fluorescence changes ($\Delta F/F_0$). The control value of $PCO_2$ was 20 mmHg and switched to 55 mmHg or 50 mM KCl at coloured bars. 3 µM ATP was applied at the end of each experiment to confirm sensor functionality. Furthermore, as a positive control, 50 mM KCl was applied to depolarise the cells and confirm channel function. (**F–G**) Box plots summarising the total ATP release in µM for each double mutant in 55 mmHg $PCO_2$ and 50 mM KCl. Data points represent individual measured cells.

The online version of this article includes the following source data for figure 6:

**Source data 1.** All the source data for the graphs in *Figure 6*.

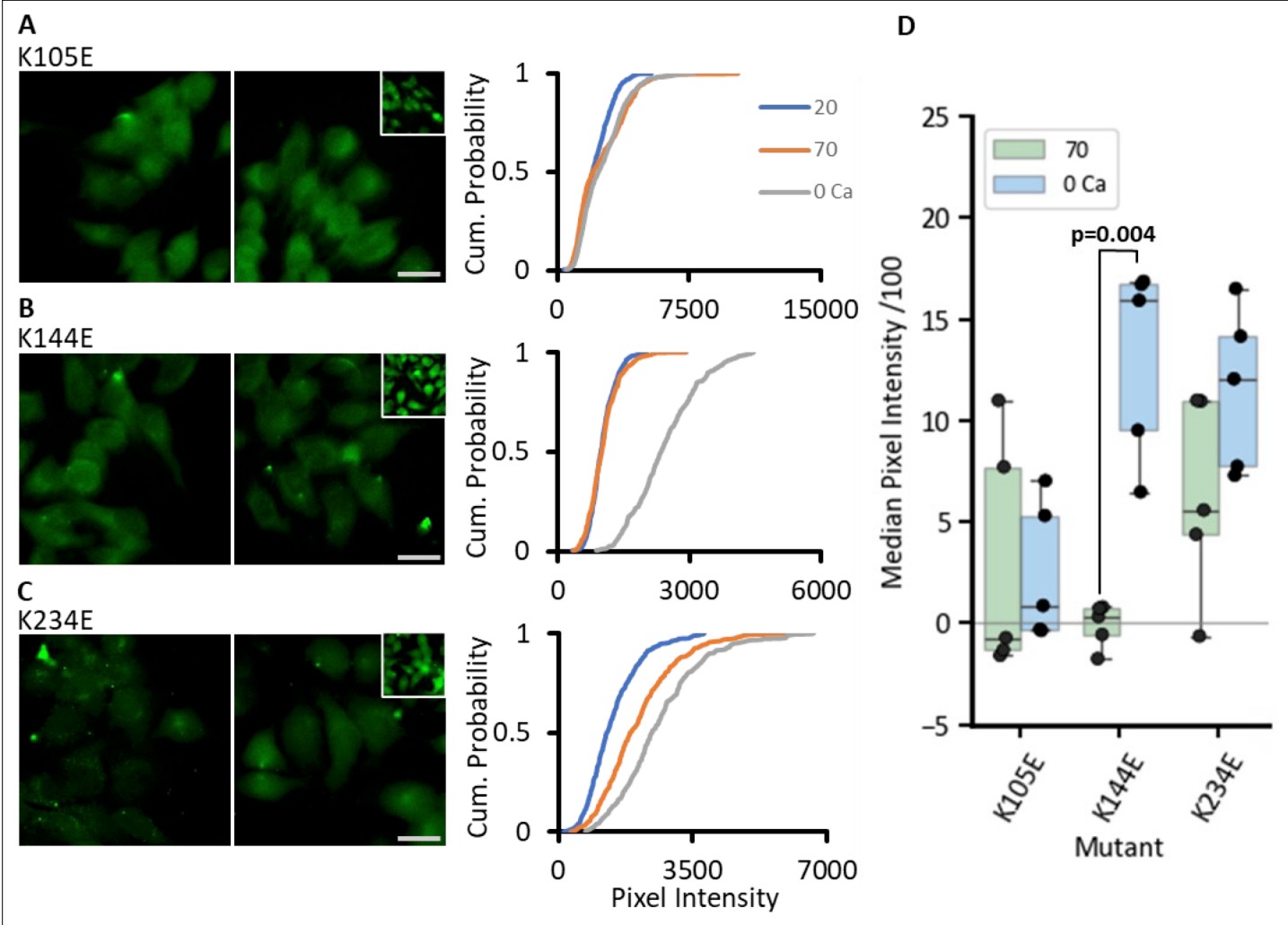

**Figure 7.** Introduction of negative charge into the carbamylation motif via Lys to Glu mutations has mixed effects on $CO_2$ sensitivity of Cx43 hemichannels. Expressing cells pixel intensity for each construct was measured from five individual transfections, with at least 40 cells per condition. (**A–C**) Representative cell images for 20 (left), 70 (right) with insets displaying the 0 $Ca^{2+}$ control for each mutant K105E (**A**), K144E (**B**), K234E (**C**). Scale bar represents 20 μm. Cumulative probability graphs of pixel intensities are shown on the right for each mutant with three conditions 20 mmHg $PCO_2$ (blue line), 70 mmHg $PCO_2$ (orange), and 0 $Ca^{2+}$ (grey line). (**D**) The box plot shows change in median pixel intensity from 20 mmHg $PCO_2$ for each transfection for 70 mmHg $PCO_2$ (green boxes) and 0 $Ca^{2+}$ (blue boxes).

The online version of this article includes the following source data for figure 7:

**Source data 1.** All the source data for the graphs in *Figure 7*.

## Pathological mutations of Cx43 cause loss of $CO_2$ sensitivity

A prominent condition associated with mutations of Cx43 is oculodental digital dysplasia (ODDD; *Laird, 2014*). This syndrome involves a range of conditions such as soft tissue fusion of the digits, abnormal craniofacial bone development, small eyes, and loss of tooth enamel. Later in age conditions such as glaucoma, skin disease and neuropathies may become evident. As pathological mutations of other $CO_2$-sensitive connexins alter their $CO_2$ sensitivity (*Meigh et al., 2014*; *de Wolf et al., 2016*; *Cook et al., 2019*; *Butler and Dale, 2023*), we studied two pathological mutations of Cx43. The mutation L90V causes ODDD (*Shibayama et al., 2005*; *Lai et al., 2006*) and A44V causes the skin condition, erythrokeratodermia variabilis et progressiva (EKVP; *Cocozzelli and White, 2019*; *Srinivas et al., 2019*). We found that the L90V mutation removed $CO_2$ sensitivity from Cx43 by both the dye loading assay and the GRAB$_{ATP}$ assay (*Figure 11*). The mutation A44V blocked $CO_2$-dependent ATP release (*Figure 11*).

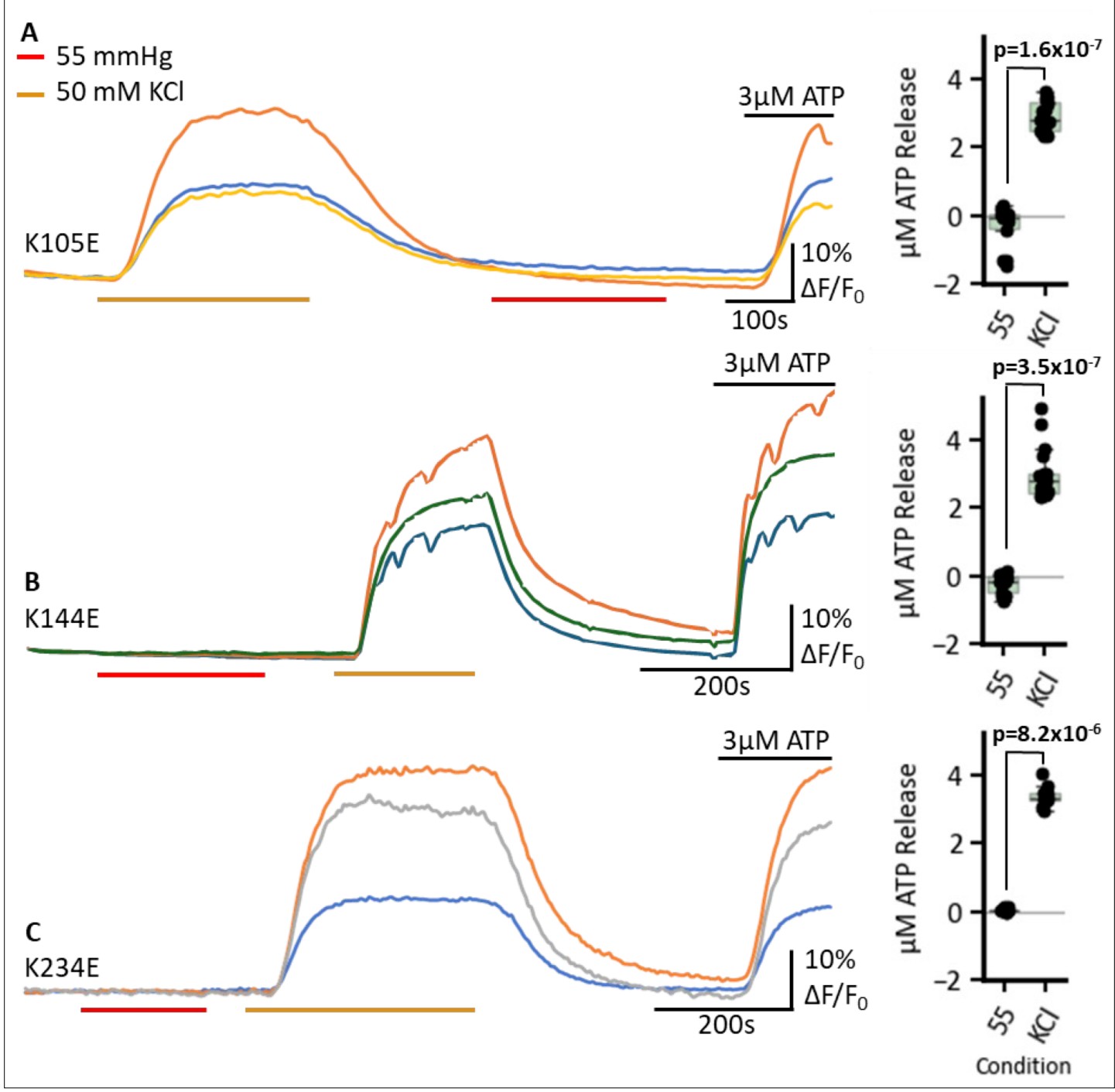

**Figure 8.** Lys to Glu mutations abolish $CO_2$-dependent ATP release. GRAB$_{ATP}$ fluorescence traces of ATP release from cells expressing single mutant connexins: K105E (**A**) n=18, K144E (**B**) n=17, and K234E (**C**) n=13, in response to 55 mmHg $PCO_2$ (from control value of 20 mmHg, red bars), 50 mM KCl depolarisation control (orange bars), and a 3 μM ATP control application at the end of all experiments to confirm sensor functionality. Traces show changes in normalised fluorescence over time (ΔF/F), indicating ATP release. On the right, the box plots display the quantified ATP release in μM for each mutant under 55 mmHg $PCO_2$ and 50 mM KCl. Each data point represents a measurement from an individual cell.

The online version of this article includes the following source data for figure 8:

**Source data 1.** All the source data for the graphs in *Figure 8*.

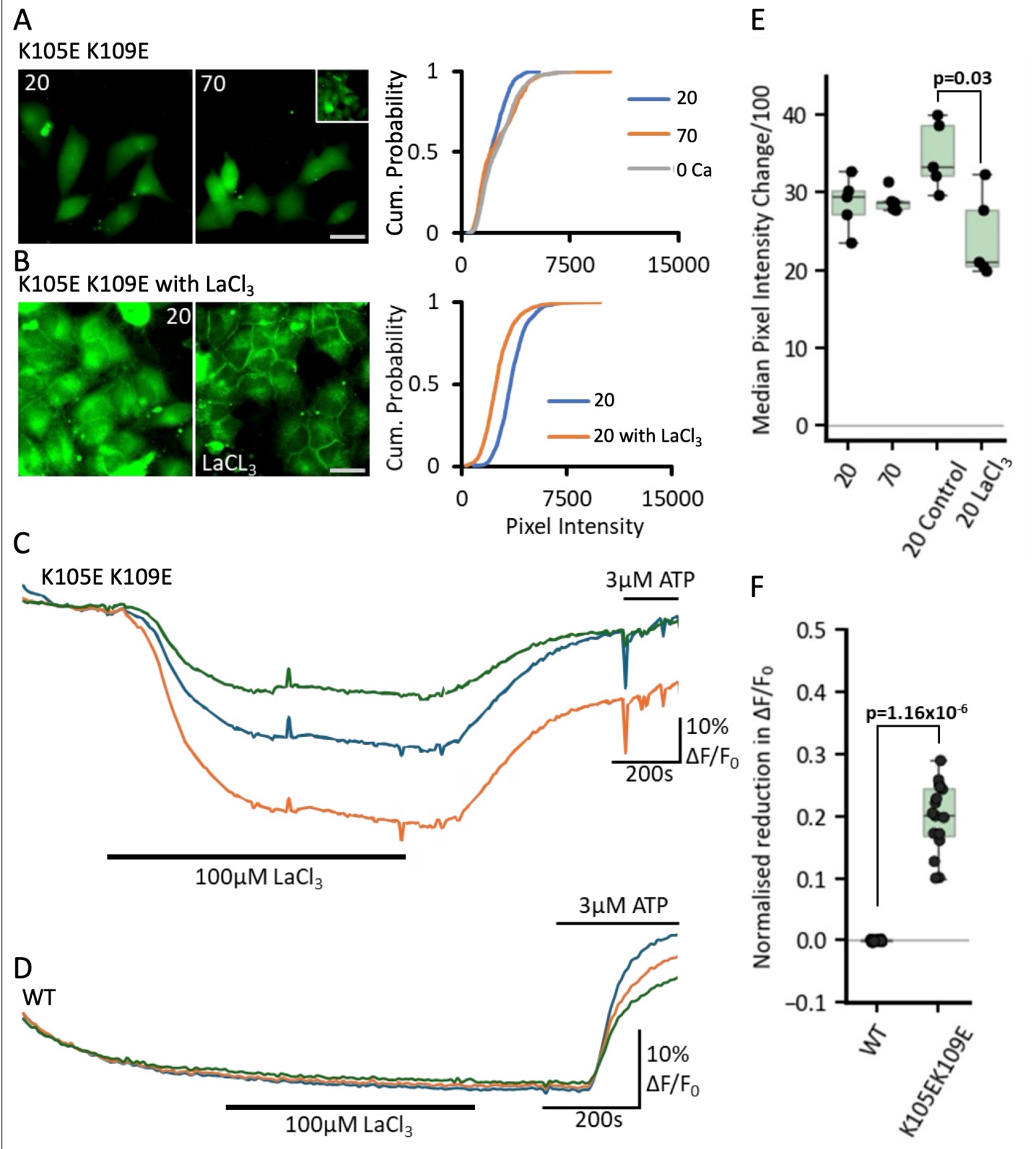

**Figure 9.** The Cx43 double mutant, K105E K109E, is constitutively open. (**A**) Representative fluorescence images of cells expressing the Cx43 K105E K109E mutant under low (20 mmHg $PCO_2$, left) and high $CO_2$ (70 mmHg $PCO_2$, right), inset shows the 0 $Ca^{2+}$ control. Scale bar is 20 µm. Cumulative probability plot of pixel intensity for each condition is shown on the right, overall indicating higher baseline fluorescence and perpetually open hemichannels. (**B**) Dye-loading with 20 mmHg $PCO_2$ (left) and 20 mmHg $PCO_2$ with 100 µm $LaCl_3$. Cumulative probability plot for pixel intensity under

*Figure 9 continued on next page*

*Figure 9 continued*

these conditions is shown on the right. (**C**) Fluorescence traces of ATP release from cells co-expressing $GRAB_{ATP}$ and Cx43 K105E K109E (n=16) under 20 mmHg $PCO_2$ and 20 mmHg $PCO_2$ with 100 µM $LaCl_3$ (black bar). Fluorescence is normalised to baseline. (**D**) Fluorescence traces of ATP release from cells co-expressing $GRAB_{ATP}$ and Cx43 WT under 20 mmHg $PCO_2$ and 20 mmHg $PCO_2$ with 100 µM $LaCl_3$ (black bar). (**E**) Box plots summarising median pixel intensity under the different conditions, showing a significant reduction in intensity in the presence of $LaCl_3$ (p=0.03). (**F**) Box plot shows normalised fluorescence changes values for the difference between – 20 mmHg $PCO_2$ (representing baseline normalisation) and the application of $LaCl_3$ for both the K105E K109E and WT Cx43 constructs.

The online version of this article includes the following video and source data for figure 9:

**Source data 1.** All the source data for the graphs in *Figure 9*.

**Figure 9—video 1.** $GRAB_{ATP}$ fluorescence recorded from HeLa cells expressing $Cx43^{K105E\ K109E}$ during application of 100 µM $LaCl_3$ in 20 mmHg $PCO_2$ aCSF.

https://elifesciences.org/articles/105989/figures#fig9video1

## Discussion

### Hemichannels of Cx43 are $CO_2$ sensitive

In contrast to Cx26 expression, which is restricted to specific, small regions of the brain (*Nagy et al., 2001*; *Rash et al., 2001*; *Nagy et al., 2011*), Cx43 serves as the principal astrocytic connexin, exhibiting widespread presence throughout the brain (*Schulz et al., 2015*; *Yin et al., 2018*; *de Ceglia et al., 2023*). This widespread distribution, combined with our results, suggests that Cx43 gives the capacity for $CO_2$ sensitivity across the entire brain.

Our data from three independent assays (dye loading, measurement of ATP release and whole cell recordings) show that hemichannels of Cx43 open to modest changes in $PCO_2$. However, Cx43 gap junction channels are insensitive to these same changes in $PCO_2$. At physiological levels of $PCO_2$, Cx43 hemichannels are partially open and able to release ATP. This observation potentially resolves a puzzling inconsistency in the literature. Many biophysical recordings of Cx43 hemichannels, performed in simplified buffered salines without appreciable dissolved $CO_2$ or $HCO_3^-$, document Cx43 hemichannels as being shut (*Contreras et al., 2003*; *Schalper et al., 2010*). By contrast, under physiological conditions both in vitro and in vivo, which necessarily involve $CO_2$-$HCO_3^-$ buffering, the evidence suggests that Cx43 hemichannels remain open to some extent (*Chever et al., 2014*; *Guillebaud et al., 2020*; *Turovsky et al., 2020*). Continual ATP release from astrocytes via Cx43 has been shown to give a tone of ATP that enhances the strength of synaptic transmission in the hippocampus by about 20% (*Chever et al., 2014*). Our data is highly complementary to these observations as we find an enhancement of fEPSP amplitude of about 30% when increasing $PCO_2$ from 20 to 35 mmHg. It is important to note that our experiments were performed under isohydric conditions and differ from a prior study that examined hypocapnia with simultaneous alkalinisation. This manipulation enhanced fEPSP strength through a reduction in extracellular adenosine and consequently less inhibition via A1 receptors (*Dulla et al., 2005*).

While our data suggests that Cx43 hemichannels are sensitive to $PCO_2$ over the range 20–70 mmHg, there is an indication that there may be some $CO_2$-dependent inhibition of hemichannel opening above 55 mmHg. Slightly less ATP release was consistently seen at a $PCO_2$ of 70 mmHg compared to 55 mmHg. Furthermore, we cannot be sure that Cx43 hemichannels are completely shut at a $PCO_2$ of 20 mmHg, as it is not possible to reduce the $PCO_2$ further without changing to HEPES-buffered salines. That Gap26 reduced synaptic transmission below the baseline amplitude recorded in slices bathed in 20 mmHg aCSF suggests that even under this condition there may still be a low level of Cx43 hemichannel gating at least in neural tissue which will continually generate $CO_2$ via oxidative phosphorylation.

Cx43 gap junction channels and hemichannels are closed by intracellular acidification. This involves interaction between the C-terminus and the cytoplasmic loop (*Duffy et al., 2002*; *Hirst-Jensen et al., 2007*). Similar interactions between the C-terminus and cytoplasmic loop are involved in regulating hemichannel activity stimulated by increases in intracellular $Ca^{2+}$ (*Iyyathurai et al., 2018*). However, the $CO_2$-dependent opening of Cx43 is independent of these mechanisms as it persists when the C-terminus has been deleted.

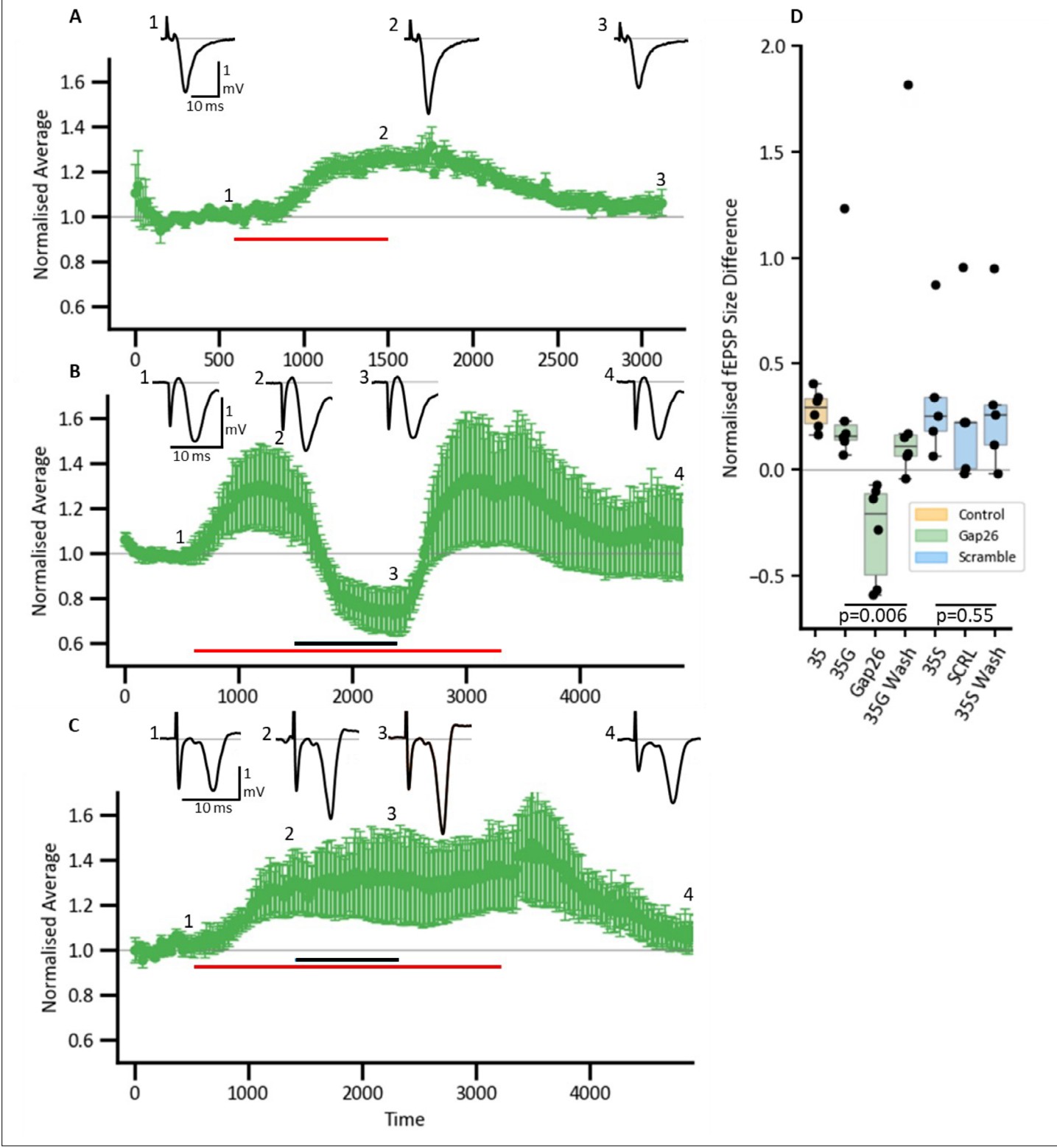

**Figure 10.** PCO$_2$-dependent modulation of amplitude of fEPSPs in hippocampus is mediated via Cx43. (**A–C**) Time-course plots showing the average amplitude of the normalised fEPSP (± SEM) amplitude in response to different conditions. Inserts display representative fEPSPs. (**A**) Control condition showing an increase in fEPSP amplitude in response to a modest change to PCO$_2$ (20–35 mmHg), red bar represents the application of 35 mmHg PCO$_2$, baseline conditions – 20 mmHg PCO$_2$. (**B**) EPSP responses in the presence of Gap26 (black bar) and subsequent wash. (**C**) EPSP responses with

*Figure 10 continued on next page*

*Figure 10 continued*

scrambled Gap26 peptide applied (black bar). (**D**) Box plots representing the normalised fEPSP size difference (baseline was subtracted) with colour coded conditions: orange – control 35 mmHg $PCO_2$, green – Gap26 with pre (35 G), after (35 S Wash).

The online version of this article includes the following source data for figure 10:

**Source data 1.** All the source data for the graphs in *Figure 10*.

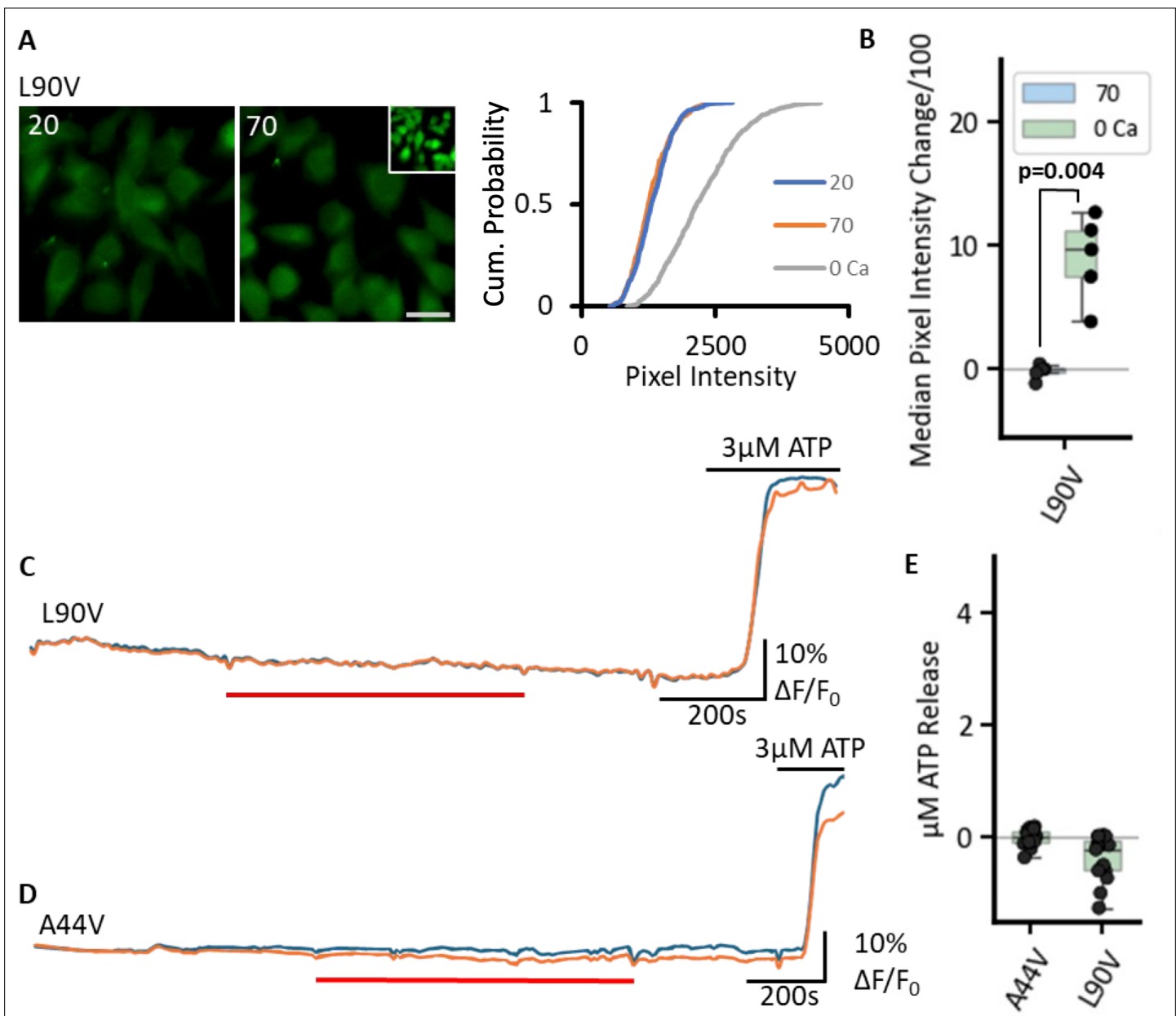

**Figure 11.** Pathological mutations of Cx43 remove its sensitivity to $CO_2$. (**A, B**) L90V prevents $CO_2$-dependent dye loading. The dye loading assay shows no change in fluorescence between 20 and 70 mmHg, yet functional channels are expressed as shown by the zero $Ca^{2+}$-positive control (inset). Box plot shows the change in median pixel intensity from 20 mmHg $PCO_2$ to 70 mmHg $PCO_2$ (green boxes) and 0 $Ca^{2+}$ (blue boxes) for each transfection. Scale bar is 20 μm. (**C–E**) $GRAB_{ATP}$ recordings show that L90V and A44V also abolish $CO_2$-dependent ATP release (baseline 20 mmHg $PCO_2$, red bar 70 mmHg $PCO_2$).

The online version of this article includes the following source data for figure 11:

**Source data 1.** All the source data for the graphs in *Figure 11*.

## The residues involved in $CO_2$ sensitivity and possible mechanisms

By exploiting the new structural data for connexins (*Myers et al., 2018*; *Flores et al., 2020*; *Lee et al., 2020*; *Yue et al., 2021*; *Brotherton et al., 2022*; *Qi et al., 2022*; *Lee et al., 2023a*; *Lee et al., 2023b*; *Qi et al., 2023*), the predictive power of AF3 and our understanding of carbamylation and $CO_2$-dependent gating of Cx26 hemichannels and GJCs (*Nijjar et al., 2021*; *Brotherton et al., 2024*), we identified a series of residues that could plausibly be involved in mediating the $CO_2$ sensitivity of Cx43: K144, the equivalent of K125 in Cx26; K105, the analogue of R104 in Cx26; K109, the equivalent of K108 in Cx26; and K234, an equivalent of R216 in Cx26. Both K125 and K108 in Cx26 are known to be carbamylated (*Nijjar et al., 2025*). Carbamylation of K125 is essential for hemichannel opening to $CO_2$, whereas carbamylation of both K125 and K108 is required for GJC closure to $CO_2$ (*Nijjar et al., 2025*). R216 in Cx26 has been tentatively identified as a possible interacting partner for K108 but this could be indirect (*Nijjar et al., 2025*). Crucially, the spatial arrangements of K144, K105, K109, and K234 in AF3 predictions indicate that these residues are sufficiently close to interact across the subunit boundary.

In Cx26, K125 is proposed to interact with R104 following carbamylation, and interestingly mutation of either K125 or R104 is sufficient to abolish $CO_2$-dependent hemichannel opening and GJC closure. Mutation of K108 is also sufficient by itself to abolish GJC closure to $CO_2$ (*Nijjar et al., 2025*). By contrast, single mutations to Gln of any of the 4 Lys residues identified in Cx43 as the equivalents to those involved in $CO_2$-dependent gating of Cx26 were unable to completely abolish $CO_2$ sensitivity. Similarly, the mutations K125E and R104E in Cx26 create constitutively open hemichannels. However, single Lys to Glu mutations in Cx43 did not have this effect. Our data show that in Cx43 at least two of the four identified residues must be mutated to completely abolish $CO_2$ sensitivity or to give constitutively open hemichannels. For example, mutation of any two Lys residues to Gln completely abolishes $CO_2$-dependent ATP release and dye loading. Conversely, the double mutation, K105E and K109E, is required for constitutively open hemichannels.

This suggests that in Cx43, there is some redundancy in the effect of $CO_2$. One possible interpretation is that two carbamate bridges are formed: one being between K144 and K105 (the direct equivalent of K125-R104 in Cx26) and the second being between K109 and K234. There is certainly some data that support this: the double mutations K109Q-K144Q, K105Q-K234Q, and K144Q-K234Q being completely insensitive to $CO_2$. These double mutations would remove both of the proposed bridges. However, our observation that the mutations K105Q-K144Q and K109Q-K234Q also completely remove $CO_2$ sensitivity is inconsistent with a simple two-bridge hypothesis, as these double mutations would only affect one of the bridges and should therefore leave the other intact to give some degree of $CO_2$ sensitivity. Instead, we favour a hypothesis in which any of these four residues could be carbamylated but that they all have potential to interact. Depending on which residues become carbamylated, a bridge could form between K105 and K234 or between K109 and K144, and possibly even K144 and K234. There is some support for this idea from AF3 as K105 and K109 are only one turn distant in an alpha helix and both could therefore interact with either K144 or K234 following carbamylation. The residues in Cx43 that are involved in $CO_2$ sensitivity are equivalent to those identified in Cx26, where they participate in both hemichannel and gap junction channel gating to $CO_2$. However, in Cx43, these residues are only involved in mediating the sensitivity of the hemichannel to $CO_2$. In this respect, it is interesting that mutations of the different Lys residues alter the dose-sensitivity of Cx43 to $CO_2$ in different ways (*Figure 4—figure supplement 1*). New cryo-EM studies of Cx43 hemichannels at vitrified different levels of $PCO_2$ would shed more light on the mechanism of opening.

## Physiological implications of the $CO_2$ sensitivity of Cx43

Cx43 is the most widely expressed connexin in the human body, being present in every organ system (*Lucaciu et al., 2023*). In metabolically highly active organs such as liver, kidney, and brain, our data suggest that the physiological $PCO_2$ will be sufficient to substantially open Cx43 hemichannels (*Hogg et al., 1984*). In the context of the brain, Cx43 is the main astrocytic connexin and is expressed in all subtypes of astrocytes (*de Ceglia et al., 2023*). This would imply that $CO_2$ sensitivity mediated via Cx43 will extend to potentially all brain regions. As astrocytes are non-excitable, they are unlikely to become sufficiently depolarised for Cx43 hemichannels to open via voltage-dependent gating. We suggest therefore, in non-excitable cells such as astrocytes, that variations in $PCO_2$ may be the most important physiological regulator of Cx43 hemichannel gating. For astrocytes, $CO_2$-dependent

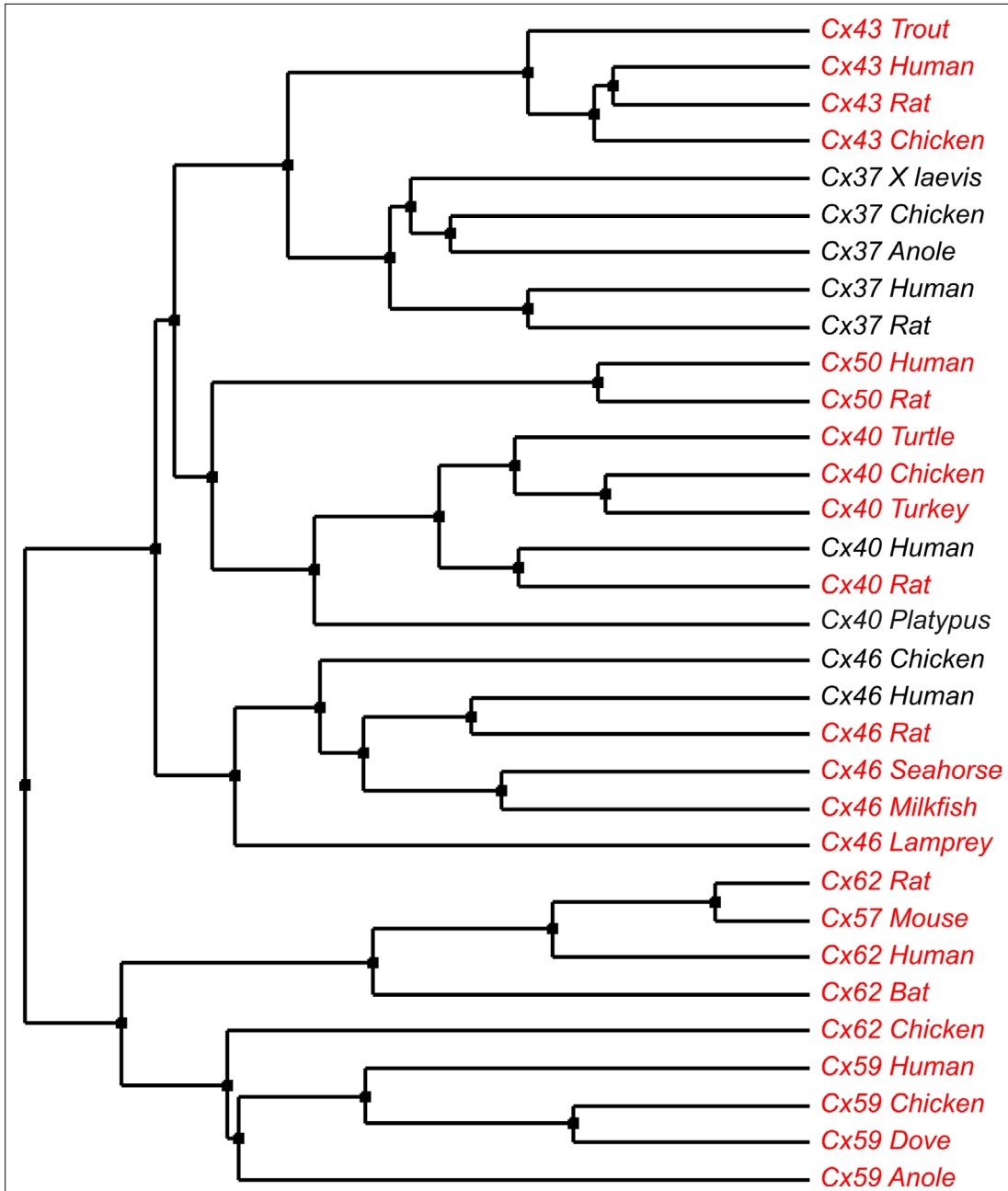

**Figure 12.** Occurrence of the carbamylation motif in the alpha connexin clade. Names in red indicate the presence of the motif. The sequences were aligned in TCoffee to check for the presence of the motif. The molecular phylogenetic tree was constructed from the aligned traces by the average distance using Blosum62 in JalView (***Waterhouse et al., 2009***).

The online version of this article includes the following source data for figure 12:

**Source data 1.** Accession numbers for the protein sequences in the tree shown in *Figure 12*.

opening of Cx43 hemichannels is likely to result in the release of a mix of small, neurochemically significant molecules such as ATP, glutamate, D-serine and lactate, depending on their intracellular concentration.

Cx43 is also highly expressed in the heart, if a proportion of the population were to be in hemichannel form as opposed to GJCs (***De Smet et al., 2021***), the newly discovered $CO_2$ sensitivity of hemichannels may have profound implications for heart function given that cardiomyocytes are highly active, and will thus generate high levels of $CO_2$ through oxidative phosphorylation. There is a

long-established link between $CO_2$ and cardiac function with effects such as increased coronary blood flow in response to $CO_2$ as well as changes in myocardial contractile function (*Crystal, 2015*). $CO_2$ can even cause arrhythmias (*Zhang et al., 2019*). The effects of $CO_2$ may be mediated through changes in intracellular pH, which is well known to alter the function of cardiac myocytes (*Spitzer et al., 2002*; *Vaughan-Jones et al., 2009*; *Orlowski et al., 2025*). However, given that Cx43 is directly sensitive to $CO_2$, an investigation as to whether $CO_2$ also has direct effects on cardiac function that are independent of consequent changes in intracellular pH seems warranted.

NF-kappa B and innate immunity exhibit sensitivity to $CO_2$ (*Cummins et al., 2010*; *Keogh et al., 2017*). Macrophages are $CO_2$ sensitive, although at least some of this is indirect and via pH changes mediated by carbonic anhydrase activity (*Strowitzki et al., 2022*). However, a direct effect of $CO_2$ on components of the immune system cannot be excluded. In cell culture, when carbonic anhydrase activity is blocked in THP-1 monocytes, the effects of $CO_2$ are significantly diminished but not abolished, suggesting the existence of a pH-independent pathway (*Strowitzki et al., 2022*). As Cx43 is expressed in monocytes it is a plausible candidate to mediate direct $CO_2$ sensitivity in these cells. Furthermore, Cx43 function in macrophages is deemed critical for various physiological and pathophysiological processes (*Rodjakovic et al., 2021*). The role of $CO_2$ on such pathways remains to be explored.

It is interesting that pathological mutations of Cx43 abrogate $CO_2$ sensitivity. This follows a pattern seen for Cx26, where a range of mutations linked to syndromic and non-syndromic hearing loss (*Meigh et al., 2014*; *de Wolf et al., 2016*; *Cook et al., 2019*), and for Cx32, where several mutations that cause X-linked Charcot Marie Tooth Disease (*Butler and Dale, 2023*) alter or completely block $CO_2$ sensitivity. Further studies are needed to see whether loss of $CO_2$ sensitivity from Cx43 does indeed contribute to pathology.

### Evolution of $CO_2$ sensitivity in connexins

Our previous findings showed that a common carbamylation motif is prevalent across the beta-connexin clade and confers $CO_2$-sensitivity on the connexins that possess it (*Dospinescu et al., 2019*). This motif for example is present in shark Cx32 (which is $CO_2$ sensitive). Humans and sharks last shared a common ancestor about 400–450 MYA. Our discovery that a very similar carbamylation motif is present in Cx43 and confers $CO_2$ sensitivity onto their hemichannels. Examining the sequences of other human and non-human alpha connexins reveals that the carbamylation motif with appropriately oriented residues is present in human Cx50, Cx59, and Cx62 as well as Cx40, Cx46, and Cx57 of non-human species (*Figure 12*, *Figure 12—source data 1*). We have shown Cx50 hemichannels are $CO_2$ sensitive and this also depends on the carbamylation motif (*Lovatt et al., 2025*). These findings suggest that the carbamylation motif and $CO_2$ sensitivity must have been present in the ancestral connexin gene that predated the establishment of the alpha and beta connexin clades. Further work is needed to understand the evolution of the carbamylation motif and $CO_2$ sensitivity, but our observations indicate that it is an ancient feature within the molecular phylogeny of the connexins, rather than being a more recently derived feature restricted to the beta connexins.

## Methods

### Key resources table

| Reagent type (species) or resource | Designation | Source or reference | Identifiers | Additional information |
|---|---|---|---|---|
| Cell line (human) | HeLa DH | UK Health Security Agency | RRID:CVCL_2483 | |
| Chemical compound, drug | DMEM | Merck Life Sciences UK Ltd | CAT# D6046 | |
| Chemical compound, drug | Fetal bovine serum | Labtech.com | CAT# FCS-SA | |
| Chemical compound, drug | GeneJuice Transfection Reagent | Merck Life Sciences UK Ltd | CAT# 70967–3 | |
| Chemical compound, drug | PEI Prime linear polyethylenimine | Merck Life Sciences UK Ltd | CAT# 919012 | |

*Continued on next page*

*Continued*

| Reagent type (species) or resource | Designation | Source or reference | Identifiers | Additional information |
|---|---|---|---|---|
| Chemical compound, drug | 5 (6)-Carboxyfluorescein | Merck Life Sciences UK Ltd | CAT# 8510820005 | |
| Chemical compound, drug | Opti-MEM I Reduced Serum Medium | Thermo Fisher Scientific | CAT# 31985070 | |
| Recombinant DNA reagent | pDisplay-GRAB_ATP1.0 plasmid | Addgene | plasmid#167582; RRID:Addgene_167582 | |
| Chemical compound, drug | GeneJuice | Merck Sigma-Aldrich | CAT# 70967 | |
| Chemical compound, drug | 5 (6)-Carboxyfluorescein | Novabiochem | CAT# 8.51082 | |
| Chemical compound, drug | 2-Deoxy-2-[(7-nitro-2,1,3-benzoxadiazol-4-yl)amino]-D-glucose | AAT Bioquest | CAT# 36702 | |
| Chemical compound, drug | BCECF, AM (2',7'-Bis-(2-Carboxyethyl)–5-(and-6)-Carboxyfluorescein, Acetoxymethyl Ester) | Thermo Fisher Scientific | CAT# 11524147 | |
| Chemical compound, drug | Gap26 and scrambled peptides | Genscript Biotech UK Ltd | | Custom synthesis |
| Sequence-based reagent | Cx43-forward | IDT | PCR primers | TACCGCGGGCCCGGGATCCACCGGTATGGGTGACTGGAGCGCC |
| Sequence-based reagent | Cx43-reverse | IDT | PCR primers | GCGGTACCCCGATCTCCAGGTCATCAGGCC |
| Sequence-based reagent | mCherry-forward | IDT | PCR primers | CCTGGAGATCGGGGTACCGCGGGCCCGG |
| Sequence-based reagent | mCherry-reverse | IDT | PCR primers | CTTGATACTTACCTGCGGCCTCGAGTTACTTGTACAGCTCGTCCATGCCGCCG |
| Sequence-based reagent | K105Q-forward | IDT | PCR primers | GGAAGAGCAACTGAACAAGAAAGAGGAAG |
| Sequence-based reagent | K105Q-reverse | IDT | PCR primers | TCTTGTTCAGTTGCTCTTCCTTTCGCATCACATAG |
| Sequence-based reagent | K109Q-forward | IDT | PCR primers | GAACAAGCAAGAGGAAGAACTCAAGGTTGCCC |
| Sequence-based reagent | K109Q-reverse | IDT | PCR primers | GTTCTTCCTCTTGCTTGTTCAGTTTCTCTTCCTTTCG |
| Sequence-based reagent | K144Q-forward | IDT | PCR primers | GCATGGTCAGGTGAAAATGCGAGGGGGG |
| Sequence-based reagent | K144Q-reverse | IDT | PCR primers | GCATTTTCACCTGACCATGCTCTTCAATACCGTAC |
| Sequence-based reagent | 234Q-forward | IDT | PCR primers | TTTCTTCCAGGGCGTTAAGGATCGGGTTAAGG |
| Sequence-based reagent | 234Q-reverse | IDT | PCR primers | CCTTAACGCCCTGGAAGAAAACATAGAAGAGTTCAATGATATTCAAG |
| Sequence-based reagent | 105E-forward | IDT | PCR primers | GGAAGAGGAACTGAACAAGAAAGAGGAAG |
| Sequence-based reagent | 105E-reverse | IDT | PCR primers | TCTTGTTCAGTTCCTCTTCCTTTCGCATCACATAG |
| Sequence-based reagent | 144E-forward | IDT | PCR primers | GCATGGTGAGGTGAAAATGCGAGGGGGG |
| Sequence-based reagent | 144E-reverse | IDT | PCR primers | GCATTTTCACCTCACCATGCTCTTCAATACCGTAC |

*Continued on next page*

*Continued*

| Reagent type (species) or resource | Designation | Source or reference | Identifiers | Additional information |
|---|---|---|---|---|
| Sequence-based reagent | 234E-forward | IDT | PCR primers | TTTCTTCGAGGGCGTTAAGGATCGGGTTAAGG |
| Sequence-based reagent | 234E-reverse | IDT | PCR primers | CCTTAACGCCCTCGAAGAAAACATAGAAGAGTTCAATGATATTCAAG |
| Sequence-based reagent | K105Q K109Q-forward | IDT | PCR primers | GCAACTGAACAAGCAAGAGGAAGAACTCAAGGTTGCCC |
| Sequence-based reagent | K105Q K109Q-reverse | IDT | PCR primers | GTTCTTCCTCTTGCTTGTTCAGTTGCTCTTCCTTTC |
| Sequence-based reagent | K105E K109E-forward | IDT | PCR primers | GGAACTGAACAAGGAAGAGGAAGAACTCAAGGTTGCCC |
| Sequence-based reagent | K105E K109E-reverse | IDT | PCR primers | GTTCTTCCTCTTCCTTGTTCAGTTCCTCTTCCTTTC |
| Sequence-based reagent | Cx43$^{1-256}$ forward | IDT | PCR primers | TTT GGC AAA GAA TTC GGT ACC GCG GGC CCG GGA TCC AC |
| Sequence-based reagent | Cx43$^{1-256}$ reverse | IDT | PCR primers | GGA TCC CGG GCC CGC GGT ACC CCT TTG GCA GGG CTC AGC GC |
| Software, algorithm | PyMol | https://pymol.org/ | RRID:SCR_000305 | |
| Software, algorithm | AlphaFold3 | https://alphafoldserver.com/ | RRID:SCR_028034 | |
| Software, algorithm | JalView 2.11.5.0 | https://www.jalview.org | RRID:SCR_006459 | |

## Recording solutions

Hypocapnic (20 mmHg $PCO_2$): 140 mM NaCl, 10 mM $NaHCO_3$, 1.25 mM $NaH_2PO_4$, 3 mM KCl, 1 mM $MgSO_4$, 10 mM D-glucose and 2 mM $CaCl_2$. This was bubbled with a mix of 95%$O_2$/5%$CO_2$ and balanced with sufficient pure $O_2$ from an oxygen concentrator to give a final pH of ~7.3.

Control (35 mmHg $PCO_2$): 124 mM NaCl, 26 mM $NaHCO_3$, 1.25 mM $NaH_2PO_4$, 3 mM KCl, 10 mM D-glucose, 1 mM $MgSO_4$, 2 mM $CaCl_2$. This was bubbled with 95%$O_2$/5% $CO_2$ and had a final pH of ~7.3.

Hypercapnic (55 mmHg $PCO_2$): 100 mM NaCl, 50 mM $NaHCO_3$, 1.25 mM $NaH_2PO_4$, 3 mM KCl, 10 mM D-glucose, 1 mM $MgSO_4$, 2 mM $CaCl_2$. This was bubbled with sufficient $CO_2$ (~9%, balance $O_2$) to give a final pH of ~7.3.

Hypercapnic (70 mmHg $PCO_2$): 73 mM NaCl, 80 mM $NaHCO_3$, 1.25 mM $NaH_2PO_4$, 3 mM KCl, 10 mM D-glucose, 1 mM $MgSO_4$, 2 mM $CaCl_2$. This was bubbled with sufficient $CO_2$ (approximately 12%, balance $O_2$) to give a final pH of ~7.3.

Zero $Ca^{2+}$: 140 mM NaCl, 10 mM $NaHCO_3$, 1.25 mM $NaH_2PO_4$, 3 mM KCl, 1 mM $MgSO_4$. On the day of recording, 10 mM D-glucose, 1 mM EGTA (Ethyleneglycol-bis(β-aminoethyl)-N,N,N',N'-tetraacetic acid) and 2 mM $MgCl_2$ was added, and the solution bubbled with 95% $O_2$/5% $CO_2$ and balanced with sufficient pure $O_2$ from an oxygen concentrator to give a final pH of ~7.3.

## Cloning and mutagenesis

Plasmids containing mutated versions of Cx43 were generated using the Gibson Assembly method (*Gibson et al., 2009*). Overlapping fragments both containing the desired mutation were PCR amplified with primers. Double mutations were cloned using successive Gibson assemblies. PCR fragments were amplified using Q5 High-Fidelity DNA Polymerase (New England Biolabs; normal PCR or site-directed mutagenesis). The overlap region generated through PCR for Gibson Assembly was 20 base pairs, and the pCAG vector was used as the backbone. All Cx43 constructs were inserted upstream of an mCherry tag, linked via a 12 AA linker (GVPRARDPPVAT). All PCR primers were purchased from Integrated DNA Technologies (IDT). For double mutations, an additional round of cloning was performed using a single mutant as the template to introduce the additional mutation. For the truncation (Cx43$^{1-256}$), we created a PCR product of the truncated version of Cx43 and fused it to the same AA linker and mCherry as previously stated to allow visualisation of cellular localisation. All constructs were confirmed by DNA sequencing (Eurofins GATC sequencing or Plasmidasaurus).

## Cell culture

HeLa DH cells (obtained directly for the study from UK Health Security Agency and authenticated by ECACC) were grown in Dulbecco's Modified Eagle Medium (DMEM), supplemented with 10% fetal bovine serum, 50 µg/mL penicillin/streptomycin. Regular testing ensured that they were free from mycoplasma infection. HeLa DH cells were used for patch clamp studies on Cx43 and for the dye loading of all Cx43 variants. For dye loading experiments, cells were seeded onto coverslips at a density of $7.5 \times 10^4$ cells per well and transiently transfected with the Cx43 constructs following the PEI Transfection Reagent protocol.

## Dye-loading

Cells expressing the construct (48–72 hr) were given an initial 5-min wash with baseline solution (artificial cerebrospinal fluid – aCSF, 20 mmHg $PCO_2$). Following this, the cells were exposed to either control solution, hypercapnic (70 mmHg $PCO_2$), or a zero $Ca^{2+}$ positive control (20 mmHg $PCO_2$) solution, all of which contained 200 µM 5 (6)-carboxyfluorescein (CBF) for 10 min. Next, cells were put into 20 mmHg $PCO_2$ solution with 200 µM CBF for 5 min to close any open channels and prevent dye loss in the next step. Then, the coverslip was washed with control solution (without CBF) for another 30 min to remove extracellular dye. For each condition, a different coverslip with HeLa cells was used. Cx43 was tagged with mCherry on the C-terminus using a short linker. Expression was verified using mCherry fluorescence. The experiments were replicated using independent transfections five times.

Following dye loading, HeLa cells were imaged by epifluorescence (Scientifica Slice Scope, Cairn Research OptoLED illumination, 60 x water Olympus immersion objective, NA 1.0, Hamamatsu ImagEM EM-CCD camera, Metafluor software). CBF was excited by a 470 nm LED, with emission captured between 504 and 543 nm. Cx43 constructs had a C-terminal mCherry tag, which was excited by a 535 nm LED and emission captured between 570 and 640 nm.

Following acquisition, analysis was performed by a person blind to the experimental condition. ImageJ (Wayne Rasband, National Institutes of Health, USA) was used to measure the extent of dye loading by drawing a region of interest (ROI) around each cell, and subsequently, the mean pixel intensity of the ROI was determined. The mean pixel intensity of a representative background ROI for each image was subtracted from each cell measurement from the same image. At least 40 cells were measured for each condition per experiment, and five repetitions of independently transfected HeLa cells were completed. The mean pixel intensities were plotted as cumulative probability distributions, and these graphs show every data point measured. To assess the effect of the 70 mmHg and zero $Ca^{2+}$ solutions on dye loading, the difference in the median pixel intensities between the 70 mmHg and 20 mmHg conditions, and the zero $Ca^{2+}$ and 20 mmHg conditions was calculated for each transfection (*Figures 3, 5, 7 and 11*).

## GRAB$_{ATP}$ recordings

Cells were transiently transfected with the pCAG-Cx43-mCherry construct and pDisplay-GRAB_ ATP1.0-IRES-mCherry-CAAX (Addgene plasmid # 167582; RRID:Addgene_167582) (*Wu et al., 2022*) 48 hr prior to imaging. Cells were perfused with control aCSF until a stable baseline was reached, before perfusion with either hypercapnic or high $K^+$ aCSF (positive control). Once a stable baseline was reached after solution change, cells were again perfused with control aCSF and when a stable baseline was reached, recordings were calibrated by direct application of 3 µM of the corresponding analyte.

All cells were imaged by epifluorescence as above. The cpGFP in GRAB$_{ATP}$ was excited by a 470 nm LED, with emission captured between 504–543 nm. As the Cx43 constructs were mCherry tagged, we selected only cells that expressed both cpGFP and mCherry for recording. GRAB$_{ATP}$ fluorescence images were acquired every 4 s. For each condition, at least 3 independent transfections were performed with at least two coverslips per transfection.

Analysis of all experiments was carried out in ImageJ. Images were opened as a stack and stabilised. ROIs were drawn around cells co-expressing both sensor and connexin. Median pixel intensity was plotted as normalised fluorescence change ($\Delta F/F_0$) over time to give traces of fluorescence change. Amount of ATP release was quantified as concentration by normalising to the $\Delta F/F_0$ caused by application of 3 µM ATP.

## Intracellular pH measurement

BCECF-AM dissolved in DMSO and Pluronic F127 and diluted in 35 mmHg aCSF for a final concentration of 2 μM. Coverslips seeded with parental HeLa DH cells were incubated in BCECF for 20 min, before being washed in 35 mmHg aCSF for 10 min. Cells were perfused with control aCSF until a stable baseline was reached, before perfusion with either 55 mmHg or 70 mmHg aCSF. Once a stable baseline was reached after solution change, cells were again perfused with control aCSF. $pH_i$ was then calibrated following the method in *James-Kracke, 1992*. All cells were imaged by epifluorescence with the BCECF excited by a 470 nm LED and emission captured between 504 and 543 nm. Measurements were obtained from three independent transfections performed with at least two coverslips per transfection.

## Patch clamp recordings

Coverslips containing non-confluent HeLa cells were placed into a perfusion chamber at room temperature and superfused with control aCSF. An Axopatch 200B amplifier was used to make whole-cell recordings from single HeLa cells. The intracellular fluid in the patch pipettes contained: K-gluconate 60 mM, Cs-gluconate 50 mM, CsCl 10 mM, TEACl 10 mM, EGTA 10 mM, $Na_2ATP$ 3 mM, $MgCl_2$ 3 mM, $CaCl_2$ 1 mM, HEPES 10 mM, sterile filtered, pH adjusted to 7.2 with KOH. An agarose salt bridge was used to avoid solution changes altering the potential of the Ag/AgCl reference electrode. All whole-cell recordings were performed at a holding potential of –50 mV. Whole-cell conductance was measured by repeated steps to –40 mV. To allow for any drift in whole-cell conductance unrelated to the $CO_2$ stimulus, the maximal conductance during the $CO_2$ test stimulus was compared to the average of the conductance before the stimulus and after the stimulus had been fully washed off. Each whole cell recording was considered to be an independent statistical replicate.

Control (35 mmHg) aCSF was used as the baseline and switched to 20 mmHg, 55 mmHg or 70 mmHg aCSF to measure the conductance responses to differing levels of $PCO_2$. To convert these responses to a $PCO_2$ dose-response curve, assigned a value of zero to 20 mmHg and plotted all changes relative to this. Thus, the values at 35 mmHg were the absolute values of the changes evoked by going from 35 to 20 mmHg, and the absolute value of median change from 35 to 20 mmHg was added to the changes observed going from 35 to 55 and 35 to 70 mmHg.

## Imaging assay of gap junction transfer

2-Deoxy-2-[(7-nitro-2,1,3-benzoxadiazol-4-yl)amino]-D-glucose, NBDG, was included at 200 μM in the patch recording fluid, which contained: K-gluconate 130 mM; KCl 10 mM; EGTA 5 mM; $CaCl_2$ 2 mM, HEPES 10 mM, pH was adjusted to 7.3 with KOH to give a resulting final osmolarity of 295 mOsm. Cells were imaged on a Cleverscope (MCI Neuroscience) with a Photometrics Prime camera under the control of Micromanager 1.4 software. LED illumination (Cairn Research) and an image splitter (Optosplit, Cairn Research) allowed simultaneous imaging of the mCherry-tagged Cx43 subunits and the diffusion of the NBDG into and between cells. Coupled cells for intercellular dye transfer experiments were initially selected based on tagged Cx43 protein expression and the presence of a gap junctional plaque (*Figure 4—figure supplement 1*). Dye permeation between cells was measured at $PCO_2$ levels of 20, 55, and 70 mmHg. After establishing the whole cell mode of recording, images were collected every 10 s. The assay was performed as described by *Nijjar et al., 2021*. The start of recording was taken to be the first image following the establishment of a stable whole cell recording. For each gap junction recording (considered as an independent statistical replicate), analysis of the cell images was performed in ImageJ using an ROI drawn on each cell to measure the median pixel intensity of the donor and acceptor cells. The time for the median fluorescence of the acceptor cell to reach 10% of the donor cell was then determined and used as the value for statistical comparisons.

## Hippocampal slice preparation

Mice of either sex, 3–5 weeks old, were killed by cervical dislocation and decapitated in accordance with United Kingdom Animals (Scientific Procedures) Act (1986) and approved by the University of Warwick AWERB and falling under the authority of licence PP7458325. The brain was dissected and kept on ice; both the cerebellum and the rostral section of the brain were removed. 400-μm-thick sagittal slices were cut with a vibratome (Microm HM 650 V microsliver) in ice-cold cutting solution composed of (in mM: 85 NaCl, 2.5 KCl, 0.5 $CaCl_2$, 1.25 $NaH_2PO_4$, 24 $NaHCO_3$, 25 glucose, and 75

sucrose; pH was adjusted to 7.4, bubbled with 95% $O_2$ and 5% $CO_2$). Then slices were incubated in aCSF composed of (in mM: 127 NaCl, 1.9 KCl, 2 $MgCl_2$, 2 $CaCl_2$, 1.2 $KH_2PO_4$, 26 $NaHCO_3$, 10 D-glucose, pH 7.4, when bubbled with 95% $O_2$ and 5% $CO_2$) at 33 °C. For fEPSP recordings, an individual slice was pre-incubated in 20 mmHg $PCO_2$ aCSF for 30 min, following which the slice was transferred to a recording chamber submerged in aCSF at 32 °C. For stimulation, pulses were delivered by a stimulator via concentric bipolar metal stimulating electrode placed on the surface of CA3. For extracellular recordings, an aCSF-filled microelectrode was placed in CA1.

## Statistics

Data is plotted as either cumulative probabilities (showing every data point) or box and whisker plots where the box is interquartile range, bar is median, and whisker extends to most extreme data point that is no more than 1.5 times the interquartile range. All individual data points are superimposed on the plots.

In the case of dye loading, one transfection was considered one replicate. In the case of patch clamping and $GRAB_{ATP}$ experiments, one cell was considered one replicate. For dye-loading, to assess the effect a mutation would have on the protein, the difference in dye loading evoked by 70 mmHg $PCO_2$ or 0 $Ca^{2+}$ conditions (from the 20 mmHg baseline) for each construct was compared, using the median value for each condition from each transfection. The Mann-Whitney U one-sided test was used for the comparison on the basis that a priori we expected mutations to either reduce $CO_2$ sensitivity or have no effect. To compare the fEPSP size across different conditions, the non-parametric Friedman two-way ANOVA was used. Statistical analysis was performed using the Python language and SciPy library.

## Acknowledgements

We thank Prof Alexander Cameron for commenting on a draft of this paper. We thank the BBSRC (BB/T013346/1, ND) for support. JB was supported by the Biotechnology and Biological Sciences Research Council (BBSRC) and University of Warwick funded Midlands Integrative Biosciences Training Partnership (MIBTP) grant number BB/T00746X/1. VMD was funded by the Medical Research Council through the University of Warwick Doctoral Training Partnership, grant number MR/N014294/1.

## Additional information

### Funding

| Funder | Grant reference number | Author |
|---|---|---|
| Medical Research Council | MR/N014294/1 | Valentin Mihai Dospinescu |
| Biotechnology and Biological Sciences Research Council | BB/T00746X/1 | Jack Butler |
| Biotechnology and Biological Sciences Research Council | BB/T013346/1 | Nicholas Dale |

The funders had no role in study design, data collection and interpretation, or the decision to submit the work for publication.

### Author contributions

Valentin Mihai Dospinescu, Data curation, Formal analysis, Investigation, Writing – original draft, Writing – review and editing; Alexander Mascarenhas, Conceptualization, Data curation, Investigation, Writing – review and editing; Jack Butler, Sarbjit Nijjar, Data curation, Investigation, Writing – review and editing; Kyara de Oliveira Taborda, Sean Connors, Investigation, Writing – review and editing; Lumei Huang, Conceptualization, Investigation, Methodology, Writing – review and editing; Nicholas Dale, Conceptualization, Data curation, Formal analysis, Supervision, Funding acquisition, Writing – original draft, Writing – review and editing

## Author ORCIDs

Valentin Mihai Dospinescu https://orcid.org/0000-0002-5988-8409
Alexander Mascarenhas https://orcid.org/0009-0008-6845-4726
Jack Butler https://orcid.org/0000-0003-2985-6841
Lumei Huang https://orcid.org/0000-0001-9479-2181
Nicholas Dale https://orcid.org/0000-0003-2196-2949

Reviewer #1 (Public review): https://doi.org/10.7554/eLife.105989.3.sa1
Reviewer #2 (Public review): https://doi.org/10.7554/eLife.105989.3.sa2
Reviewer #3 (Public review): https://doi.org/10.7554/eLife.105989.3.sa3
Author response https://doi.org/10.7554/eLife.105989.3.sa4

## Additional files

### Supplementary files

MDAR checklist

### Data availability

All data generated in the paper is available as supplements to the figures.

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
