## [Editor Report · eLife Assessment]

This **important** study reveals that connexin43 (Cx43) hemichannels are directly activated by CO₂ through a conserved carbamylation motif, extending a mechanism previously described for β-connexins to α-connexins. The evidence is **convincing**, supported by complementary biochemical and electrophysiological analyses showing CO₂-induced hemichannel opening and ATP release in cultured cells and hippocampal slices. These findings advance our understanding of connexin regulation by metabolic gases and will be of broad interest to researchers studying cell communication, neural signaling, and gasotransmitter biology.

---

## [Referee Report · Reviewer #1 (Public review)]

Summary:

This study builds on previous work demonstrating that several beta connexins (Cx26, Cx30 and Cx32) have a carbamylation motif which renders them sensitive to CO2. In response to CO2, hemichannels composed of these connexins open, enabling diffusion of small molecules (such as ATP) between the cytosol and extracellular environment. Here, the authors have identified that an alpha connexin, Cx43, also contains a carbamylation motif, and they demonstrate that CO2 opens Cx43 hemichannels. Most of the study involves using transfected cells expressing wild-type and mutant Cx43 to define amino acids required for CO2 sensitivity. Hippocampal tissue slices in culture were used to show that CO2-induced synaptic transmission was affected by Cx43 hemichannels, providing a physiological context. The authors point out that the Cx43 gene significantly diverges from the beta connexins that are CO2 sensitive, suggesting that the conserved carbamylation motif was present before the alpha and beta connexin genes diverged.

Strengths:

The molecular analysis defining the amino acids which contribute to the CO2 sensitivity of Cx43 is a major strength of the study. The rigor of analysis was strengthened by using three independent assays for hemichannel opening: dye uptake, patch clamp channel measurements and ATP secretion. The resulting analysis identified key lysines in Cx43 that were required for CO2-mediated hemichannel opening. A double K to E Cx43 mutant produced a construct that produced hemichannels that were constitutively open, which further strengthened the analysis.

Using hippocampal tissue sections to demonstrate that CO2 can influence field excitatory postsynaptic potentials (fEPSPs) provides a native context for CO2 regulation of Cx43 hemichannels. Cx43 mutations associated with Oculodentodigital Dysplasia (ODDD) inhibited CO2-induced hemichannel opening, although the mechanism by which this occurs was not elucidated.

Cytosolic pH was measured and it was further demonstrated that Cx43 hemichannels composed of untagged Cx43 are sensitive to CO2.

A molecular phylogenetic survey was performed which identified several other non-beta connexins that have a putative carbamylation motif. How this relates to connexin evolution was added to the discussion.

Weaknesses:

Cultured cells are typically grown in incubators containing 5% CO2 which is ~40 mmHg. Determining compensatory mechanisms that enable the cells to be viable if Cx43 hemichannels are open at this PCO2 would strengthen the study.

Experiments using Gap26 to inhibit Cx43 hemichannels in fEPSP measurements used a scrambled peptide as a control. Including gap peptides specifically targeting Cx26, Cx30 and Cx32 as additional controls would strengthen the study, since the tissue sections have a complex pattern of connexin expression.

---

## [Referee Report · Reviewer #2 (Public review)]

Summary:

This paper examines the CO2 sensitivity of Cx43 hemichannels and gap junctional channels in transiently transfected Hela cells using several different assays including ethidium dye uptake, ATP release, whole cell patch clamp recordings and an imaging assay of gap junctional dye transfer. The results show that raising pCO2 from 20 to 70 mmHg (at a constant pH of 7.3) cause an increase in opening of Cx43 hemichannels but did not block Cx43 gap junctions. This study also showed that raising pCO2 from 20 to 35 mm Hg resulted in an increase in synaptic strength in hippocampal rat brain slices, presumably due to downstream ATP release, suggesting that the CO2 sensitivity of Cx43 may be physiologically relevant. As a further test of the physiological relevance of the CO2 sensitivity of Cx43, it was shown that two pathological mutations of Cx43 that are associated with ODDD caused loss of Cx43 CO2-sensitivity. Cx43 has a potential carbamylation motif that is homologous to the motif in Cx26. To understand the structural changes involved in CO2 sensitivity, a number of mutations were made in Cx43 sites thought to be the equivalent of those known to be involved in the CO2 sensitivity of Cx26 and the CO2 sensitivity of these mutants was investigated.

Strengths:

This study shows that the apparent lack of functional Cx43 hemichannels observed in a number of previous in vitro function studies may be due to the use of HEPES to buffer the external pH. When Cx43 hemichannels were studied in external solutions in which CO2/bicarbonate was used to buffer pH instead of HEPES, Cx43 hemichannels showed significantly higher levels of dye uptake, ATP release, and ionic conductance. These findings may have major physiological implications since Cx43 hemichannels are found in many organs throughout the body including the brain, heart and immune system.

Weaknesses:

Interpretation of the site-directed mutation studies is complicated. Although Cx43 has a potential carbamylation motif that is homologous to the motif in Cx26, the results of site-directed mutation studies were inconsistent with a simple model in which K144 and K105 interact following carbamylation to cause the opening of Cx43 hemichannels.

Secondly, although it is shown that two Cx43 ODDD associated mutations show a loss of CO2 sensitivity, there is no evidence that the absence of CO2 sensitivity is involved in the pathology of ODDD.

---

## [Referee Report · Reviewer #3 (Public review)]

In this paper, authors aimed to investigate carbamylation effects on the function of Cx43-based hemichannels. Such effects have previously been characterized for other connexins, e.g. for Cx26, which display increased hemichannel (HC) opening and closure of gap junction channels upon exposure to increased CO2 partial pressure (accompanied by increased bicarbonate to keep pH constant). The authors used HeLa cells transiently transfected with Cx43 to investigate CO2-dependent carbamylation effects on Cx43 HC function. In contrast to Cx43-based gap junction channels that are here reported to be insensitive to PCO2 alterations, they provide evidence that Cx43 HC opening is highly dependent on the PCO2 pressure in the bath solution, over a range of 20 up to 70 mmHg encompassing the physiologically normal resting level of around 40 mmHg. They furthermore identified several Cx43 residues involved in Cx43 HC sensitivity to PCO2: K105, K109, K144 & K234; mutation of 2 or more of these AAs is necessary to abolish CO2 sensitivity. The subject is interesting and the results indicate that a fraction of HCs is open at a physiological 40 mmHg PCO2, which differs from the situation under HEPES buffered solutions where HCs are mostly closed under resting conditions. The mechanism of HC opening with CO2 gassing is linked to carbamylation and authors pinpointed several Lys residues involved in this process. Overall, the work is interesting as it shows that Cx43 HCs have a significant open probability under resting conditions of physiological levels of CO2 gassing, probably applicable to/relevant for brain, heart and other Cx43 expressing organs. The paper gives a detailed account on various experiments performed (dye uptake, electrophysiology, ATP release to assess HC function) and results concluded from those. They further consider many candidate carbamylation sites by mutating them to negatively charged Glu residues. The paper finalizes with hippocampal slice work showing evidence for connexin-dependent increases of the EPSP amplitude that could be inhibited by HC inhibition with Gap26 (Fig. 10). Another line of evidence comes from the Cx43-linked ODDD genetic disease whereby L90V as well as the A44V mutations of Cx43 prevented the CO2 induced hemichannel opening response (Fig. 11). Although the paper is interesting, in its present state it suffers from (i) a problematic Fig. 3, precluding interpretation of the data shown, and (ii) the poor use of hemichannel inhibitors that are necessary to strengthen the evidence in the crucial experiment of Fig. 2 and others.

Comments on revisions:

The traces in Fig.2B show that the HC current is inward at 20 mmHg PCO2, while it switches to an outward current at 55mmHg PCO2. HCs are non-selective channels, so their current should switch direction around 0 mV but not around -50 mV. As such, the -50 mV switching point indicates involvement of another channel distinct from non-selective Cx43 hemichannels. In the revised version, this problem has not been solved nor addressed. Additionally, I identified another problem in that the experimental traces shown lack a trace at the baseline condition of PCO2 35mmHg, while the summary graph depicts a data point. Not showing a trace at baseline PCO2 35mmHg renders data interpretation in the summary graph questionable.

---

## [Author Response]

The following is the authors’ response to the original reviews.

**Reviewer #1 (Public review):**
Summary:This study builds on previous work demonstrating that several beta connexins (Cx26, Cx30, and Cx32) have a carbamylation motif which renders them sensitive to CO_2_. In response to CO_2_, hemichannels composed of these connexins open, enabling diffusion of small molecules (such as ATP) between the cytosol and extracellular environment. Here, the authors have identified that an alpha connexin, Cx43, also contains a carbamylation motif, and they demonstrate that CO_2_ opens Cx43 hemichannels. Most of the study involves using transfected cells expressing wildtype and mutant Cx43 to define amino acids required for CO_2_ sensitivity. Hippocampal tissue slices in culture were used to show that CO_2_-induced synaptic transmission was affected by Cx43 hemichannels, providing a physiological context. The authors point out that the Cx43 gene significantly diverges from the beta connexins that are CO_2_ sensitive, suggesting that the conserved carbamylation motif was present before the alpha and beta connexin genes diverged.Strengths:(1) The molecular analysis defining the amino acids that contribute to the CO_2_ sensitivity of Cx43 is a major strength of the study. The rigor of analysis was strengthened by using three independent assays for hemichannel opening: dye uptake, patch clamp channel measurements, and ATP secretion. The resulting analysis identified key lysines in Cx43 that were required for CO_2_-mediated hemichannel opening. A double K to E Cx43 mutant produced a construct that produced hemichannels that were constitutively open, which further strengthened the analysis.(2) Using hippocampal tissue sections to demonstrate that CO_2_ can influence field excitatory postsynaptic potentials (fEPSPs) provides a native context for CO_2_ regulation of Cx43 hemichannels. Cx43 mutations associated with Oculodentodigital Dysplasia (ODDD) inhibited CO_2_-induced hemichannel opening, although the mechanism by which this occurs was not elucidated.Weaknesses:(1) Cx43 channels are sensitive to cytosolic pH, which will be affected by CO_2_. Cytosolic pH was not measured, and how this affects CO_2_-induced Cx43 hemichannel activity was not addressed.

We have now addressed this with intracellular pH measurements and removal of the C-terminal pH sensor from Cx43 -the hemichannel remains CO_2_ sensitive.

(2) Cultured cells are typically grown in incubators containing 5% CO_2_, which is ~40 mmHg. It is unclear how cells would be viable if Cx43 hemichannels are open at this PCO2.

The cells look completely healthy with normal morphology and no sign of excessive cell death in the cultures. Presumably they have ways of compensating for the effects of partially open Cx43 hemichannels.

(3) Experiments using Gap26 to inhibit Cx43 hemichannels in fEPSP measurements used a scrambled peptide as a control. Analysis should also include Gap peptides specifically targeting Cx26, Cx30, and Cx32 as additional controls.

We don’t feel this is necessary given the extensive prior literature in hippocampus showing the effect of ATP release via open Cx43 hemichannels on fEPSP amplitude that used astrocytic specific knockout of Cx43 and Gap26 (doi: 10.1523/jneurosci.0015-14.2014).

(4) The mechanism by which ODDD mutations impair CO2-mediated hemichannel opening was not addressed. Also, the potential roles for inhibiting Cx43 hemichannels in the pathology of ODDD are unclear.

These pathological mutations that alter CO_2_ sensitivity are similar to pathological mutation in Cx26 and Cx32, which also remove CO_2_ sensitivity. Our cryo-EM studies on Cx26 give clues as to why these mutations have this effect -they alter conformational mobility of the channel (Brotherton et al 2022 doi: 10.1016/j.str.2022.02.010 and Brotherton et al 2024 doi: 10.7554/eLife.93686). We assume that similar considerations apply to Cx43, but this requires improved cryoEM structures of Cx43 hemichannels at differing levels of PCO_2_.

We agree that the link between loss of CO_2_ sensitivity of Cx43 and ODDD is not established and have revised the text to make this clear.

(5) CO2 has no effect on Cx43-mediated gap junctional communication as opposed to Cx26 gap junctions, which are inhibited by CO2. The molecular basis for this difference was not determined.

Cx26 gap junction channels are so far unique amongst CO_2_ sensitive connexins in being closed by CO_2_. We have addressed the mechanism by which this occurs in Nijjar et al 2025 DOI: 10.1113/JP285885 -the requirement of carbamylation of K108 in Cx26 (in addition to K125) for GJC closure.

(6) Whether there are other non-beta connexins that have a putative carbamylation motif was not addressed. Additional discussion/analysis of how the evolutionary trajectory for Cx43 maintaining a carbamylation motif is unique for non-beta connexins would strengthen the study.

We have performed a molecular phylogenetic survey to show that the carbamylation motif occurs across the alpha connexin clade and have shown that Cx50 is indeed CO_2_ sensitive (doi: 10.1101/2025.01.23.634273). This is now in Fig 12.

**Reviewer #2 (Public review):**
Summary:This paper examines the CO_2_ sensitivity of Cx43 hemichannels and gap junctional channels in transiently transfected Hela cells using several different assays, including ethidium dye uptake, ATP release, whole cell patch clamp recordings, and an imaging assay of gap junctional dye transfer. The results show that raising pCO_2_ from 20 to 70 mmHg (at a constant pH of 7.3) causes an increase in opening of Cx43 hemichannels but does not block Cx43 gap junctions. This study also showed that raising pCO_2_ from 20 to 35 mm Hg resulted in an increase in synaptic strength in hippocampal rat brain slices, presumably due to downstream ATP release, suggesting that the CO_2_ sensitivity of Cx43 may be physiologically relevant. As a further test of the physiological relevance of the CO_2_ sensitivity of Cx43, it was shown that two pathological mutations of Cx43 that are associated with ODDD caused loss of Cx43 CO_2_-sensitivity. Cx43 has a potential carbamylation motif that is homologous to the motif in Cx26. To understand the structural changes involved in CO_2_ sensitivity, a number of mutations were made in Cx43 sites thought to be the equivalent of those known to be involved in the CO_2_ sensitivity of Cx26, and the CO_2_ sensitivity of these mutants was investigated.Strengths:This study shows that the apparent lack of functional Cx43 hemichannels observed in a number of previous in vitro function studies may be due to the use of HEPES to buffer the external pH. When Cx43 hemichannels were studied in external solutions in which CO_2_/bicarbonate was used to buffer pH instead of HEPES, Cx43 hemichannels showed significantly higher levels of dye uptake, ATP release, and ionic conductance. These findings may have major physiological implications since Cx43 hemichannels are found in many organs throughout the body, including the brain, heart, and immune system.Weaknesses:(1) Interpretation of the site-directed mutation studies is complicated. Although Cx43 has a potential carbamylation motif that is homologous to the motif in Cx26, the results of site-directed mutation studies were inconsistent with a simple model in which K144 and K105 interact following carbamylation to cause the opening of Cx43 hemichannels.

The mechanism of opening of Cx43 is more complex than that of Cx26, Cx32 and Cx50 and involves more Lys residues. The 4 Lys residues in Cx43 that are involved in opening the hemichannel have their equivalents in Cx26, but in Cx26 these additional residues seem to be involved in the closing of the GJC rather than opening of the hemichannel (see above). Cx50 is simpler and involves only two Lys residues (doi: 10.1101/2025.01.23.634273), which are equivalent to those in Cx26.

(2) Secondly, although it is shown that two Cx43 ODDD-associated mutations show a loss of CO_2_ sensitivity, there is no evidence that the absence of CO2 sensitivity is involved in the pathology of ODD

We agree, but this is probably because this has not been directly tested by experiment, as the CO_2_ sensitivity of Cx43 was not previously known. As mentioned above we have revised the text to ensure that this is clear.

**Reviewer #3 (Public review):**
In this paper, the authors aimed to investigate carbamylation effects on the function of Cx43-based hemichannels. Such effects have previously been characterized for other connexins, e.g., for Cx26, which display increased hemichannel (HC) opening and closure of gap junction channels upon exposure to increased CO_2_ partial pressure (accompanied by increased bicarbonate to keep pH constant).The authors used HeLa cells transiently transfected with Cx43 to investigate CO_2_ dependent carbamylation effects on Cx43 HC function. In contrast to Cx43-based gap junction channels that are reported here to be insensitive to PCO_2_ alterations, they provide evidence that Cx43 HC opening is highly dependent on the PCO2 pressure in the bath solution, over a range of 20 up to 70 mmHg encompassing the physiologically normal resting level of around 40 mmHg. They furthermore identified several Cx43 residues involved in Cx43 HC sensitivity to PCO2: K105, K109, K144 & K234; mutation of 2 or more of these AAs is necessary to abolish CO_2_ sensitivity. The subject is interesting and the results indicate that a fraction of HCs is open at a physiological 40 mmHg PCO_2_, which differs from the situation under HEPES buffered solutions where HCs are mostly closed under resting conditions. The mechanism of HC opening with CO_2_ gassing is linked to carbamylation, and the authors pinpointed several Lys residues involved in this process.Overall, the work is interesting as it shows that Cx43 HCs have a significant open probability under resting conditions of physiological levels of CO_2_ gassing, probably applicable to the brain, heart, and other Cx43 expressing organs. The paper gives a detailed account of various experiments performed (dye uptake, electrophysiology, ATP release to assess HC function) and results concluded from those. They further consider many candidate carbamylation sites by mutating them to negatively charged Glu residues. The paper ends with hippocampal slice work showing evidence for connexin-dependent increases of the EPSP amplitude that could be inhibited by HC inhibition with Gap26 (Figure 10). Another line of evidence comes from the Cx43-linked ODDD genetic disease, whereby L90V as well as the A44V mutations of Cx43 prevented the CO_2_-induced hemichannel opening response (Figure 11). Although the paper is interesting, in its present state, it suffers from (i) a problematic Figure 3, precluding interpretation of the data shown, and (ii) the poor use of hemichannel inhibitors that are necessary to strengthen the evidence in the crucial experiment of Figure 2 and others.

The panels in Figure 3 were mislabelled in the accompanying legend possibly leading to some confusion. This has now been corrected.

We disagree that hemichannel blockers are needed to strengthen the evidence in Figure 2 and other figures. Our controls show that the CO_2_-sensitive responses absolutely requires expression of Cx43 and was modified by mutations of Cx43. It is hard to see how this evidence would be strengthened by use of peptide inhibitors or other blockers of hemichannels that may not be completely selective.

**Reviewing Editor Comments:**
(1) Improve electrophysiological evidence, addressing concerns about the initial experiment and including peptide inhibitor data where applicable.

We think the concerns about the electrophysiological evidence arise from a misunderstanding because we gave insufficient information about how we conducted the experiments. We have now provided a much more complete legend, added explanations in the text and given more detail in the Methods. We further respond to the reviewer below.

We do not agree on the necessity of the peptide inhibitor to demonstrate dependence on Cx43. We have shown that parental HeLa cells do not release ATP to changes in PCO_2_ or voltage (Fig 2D; Butler & Dale 2023, 10.3389/fncel.2023.1330983; Lovatt et al 2025, 10.1101/2025.03.12.642803, 10.1101/2025.01.23.634273). Our previous papers have shown many times that parental HeLa cells do not load with dye to CO_2_ or zero Ca^2+^ (e.g. Huckstepp et al 2010, 10.1113/jphysiol.2010.192096; Meigh et al 2013, 10.7554/eLife.01213; Meigh et al 2014, 10.7554/eLife.04249), and we have shown that parental HeLa cells do not exhibit the same CO_2_ dependent change in whole cell conductance that the Cx43-expressing cells do (Fig 2B). In addition, we shown that mutating key residues in Cx43 alters both CO_2_-sensitive release of ATP and the CO_2_-dependent dye loading without affecting the respective positive control. To bolster this, we have included data for the K144R mutation as a supplement to Fig 3. Given the expense of Gap26 it is impractical to include this as a standard control and unnecessary given the comprehensive controls outlined.

Collectively, these data show that the responses to CO_2_ require expression of Cx43 and can be modified by mutation of Cx43.

(2) Strengthen the manuscript by measuring the effects of CO on cytosolic pH and Cx43 hemichannel opening. Consider using tail truncation mutants to assess the role of the C-terminal pH sensor in CO-mediated channel opening.

We agree and have performed the suggested experiments to address this issue.

(3) Investigate the effect of expressing the K105E/K109E Cx43 double mutant on cell viability.

In our experiments the cells look completely healthy based on their morphology in brightfield microscopy and growth rates.

(4) Discuss and analyze the uniqueness of Cx43 among alpha connexins in maintaining the carbamylation motif.

now discuss this -Cx43 is not unique. We have added a molecular phylogenetic survey of the alpha connexin clade in Fig 12. Apart from Cx37, the carbamylation motif appears in all the other members of the clade (but not necessarily in the human orthologue). In a different MS, currently posted on bioRxiv, we have documented the CO_2_ sensitivity of Cx50 and its dependence on the motif.

(5) Consider omitting data on ODDD-associated mutations unless there is evidence linking CO_2_ sensitivity to disease pathology.

This experiment is observational, and we are not making claims that there is a direct causal link. Removing the ODDD mutant findings would lose potentially useful information for anyone studying how these mutations alter channel function. We have reworded the text to ensure that we say that the link between loss of CO_2_ sensitivity and ODDD remains unproven.

(6) Justify the choice of high K^⁺^ and low external calcium as a positive control in ATP release experiments.

These two manipulations can open the hemichannel independently of the CO_2_ stimulus. Extracellular Ca^2+^ is well known to block all connexin hemichannels, and Cx43 is known to be voltage sensitive. The depolarisation from high K^+^ is effective at opening the hemichannel and we preferred this as a more physiological way of opening the Cx43 hemichannel. We have added some explanatory text.

(7) Clarify whether Cx43A44V or Cx43L90V mutations block gap junctional coupling.

This is an interesting point. Since Cx43 GJCs are not CO_2_ sensitive we feel this is beyond the scope of our paper.

(8) Discuss the potential implications of pCO₂ changes on myocardial function through alterations in intracellular pH.

We have modified the discussion to consider this point.

**Reviewer #1 (Recommendations for the authors):**
(1) Measurements of the effects of CO_2_ on cytosolic pH/Cx43 hemichannel opening would strengthen the manuscript. Since the pH sensor of Cx43 is on the C terminus, the authors could consider making tail truncation mutants to see how this affects CO_2_-mediated Cx43 channel opening.

We have done this (truncating after residue 256) -the channel remains highly CO_2_ and voltage sensitive. We have also documented the effect of the hypercapnic solutions on intracellular pH measured with BCECF. These new data are now included as figure supplements to Figure 2.

(2) What is the impact of expressing the K105E / K109E Cx43 double mutant on cell viability?

There was no obvious observed impact, cell density was as expected (no evidence of increased cell death), brightfield and fluorescence visualisation indicated normal healthy cells. We have added a movie (Fig 9, movie supplement 1) to show the effect of La^3+^ on the GRAB_ATP_ signal in cells expressing Cx43^K105E, K109E^ so readers can appreciate the morphology and its stability during the recording.

(3) A quick look at other alpha connexins suggested that Cx43 was unique among alpha connexins in maintaining the carbamylation motif. This merits additional discussion/ analysis.

This is an interesting point. Cx43 is not unique in the alpha clade in having the carbamylation motif as a number of other human alpha connexins also possess: Cx50, Cx59 and Cx62, and non-human alpha connexins (Cx40, Cx59, Cx46) also possess the motif. We have shown that Cx50 is CO_2_ sensitive. We have performed a brief molecular phylogenetic analysis of the alpha connexon clade to highlight the occurrence of the carbamylation motif. This is now presented as Fig 12 to go with the accompanying discussion.

(4) There were some minor writing issues that should be addressed. For instance, fEPSP is not defined. Also, insets showing positive controls in some experiments were not described in the figure legends.

We have corrected these issues.

**Reviewer #2 (Recommendations for the authors):**
(1) I would omit the data on the ODDD-associated mutations since there is no evidence that loss of CO_2_ sensitivity plays an important role in the underlying disease pathology.

We are not making the claim CO_2_ loss leads to the underlying pathology and have reviewed the text to ensure that we clearly express that this is a correlation not a cause. We think this is worth retaining as many pathological mutations in other CO_2_ sensitive connexins (Cx26, Cx32 and Cx50) cause loss of CO_2_ sensitivity, and this information may be helpful to other researchers.

(2) Why is high K+ rather than low external calcium used as a positive control in ATP release experiments?

We used of high K^+^ and depolarisation as a positive control as regard this as a more physiological stimulus than the low external Ca^2+^.

(3) Does Cx43A44V or Cx43L90V block gap junctional coupling?

An interesting question but we have not examined this.

(4) Provide references for biophysical recordings of Cx43 hemichannels performed in HEPES-buffered salines, which document Cx43 hemichannels as being shut.

have added the original and some later references which examine Cx43 hemichannel gating in HEPES buffer and shows the need for substantial depolarisation to induce channel opening.

(5) In the heart muscle, changes in PCO_2_ have long been hypothesized to cause changes in myocardial function by changing pHi.

This is true and we now add some discussion of this point. Now that we know that Cx43 is directly sensitive to CO_2_ a direct action of CO_2_ cannot be ruled out and careful experimentation is required to test this possibility.

**Reviewer #3 (Recommendations for the authors):**
(1) Page 3: "... homologs of K125 and R104 ... ": the context is linked to Cx26, so Cx26 needs to be added here.

Done

(2) Page 4 text and related Figure 2:(a) Figure 2A&B: PCO2-dependent Cx43 HC opening is clearly present in the carboxy-fluorescein dye uptake experiments (Figure 2A) as well as in the electrophysiological experiments (Figure 2B). The curves look quite different between these two distinct readouts: dye uptake doubles from 20 to 70 mmHg in Figure 2A while the electrophysiological data double from 45 to 70 mmHg in Figure 2B. These responses look quite distinct and may be linked to a non-linearity of the dye uptake assay or a problem in the electrophysiological measurements of Figure 2B discussed in the next point.

Different molecules/ions may have different permeabilities through the channel, which could explain the observed difference. Also, there is some contamination of the whole cell conductance change with another conductance (evident in recordings from parental HeLa cells). This is evident particularly at 70 mmHg. If this contaminating conductance were subtracted from the total conductance in the Cx43 expressing cells, then the dose response relations would be more similar. However, we are reluctant to add this additional data processing step to the paper.

(b) The traces in Figure 2B show that the HC current is inward at 20 mmHg PCO2, while it switches to an outward current at 55mmHg PCO2. HCs are non-selective channels, so their current should switch direction around 0 mV but not at -50 mV. As such, the -50 mV switching point indicates involvement of another channel distinct from non-selective Cx43 hemichannels.

We think that our incomplete description in the legend led to this misunderstanding. We used a baseline of 35 mmHg (where the channels will be slightly open) and changed to 20 mmHg to close them (or to higher PCO_2_ to open them from this baseline), hence a decrease in conductance and loss of outward current for 20 mmHg. The holding potential for the recordings and voltage steps were the same in all recordings. We have now edited the legend and added more information into the methods to clarify this and how we constructed the dose response curve.

We agree that Cx43 hemichannels are relatively nonselective and would normally be expected to have a reversal potential around 0 mV, but we are using K-Gluconate and the lowered reversal potential (~-65 mV) is likely due to poor permeation of this anion via Cx43.

(c) A Hill slope of 6 is reported for this curve, which is extremely steep. The paper does not provide any further consideration, making this an isolated statement without any theoretical framework to understand the present finding in such context (i.e., in relation to the PCO2 dependency of Cx channels).

Yes, we agree -it seems to be the case with all CO_2_ sensitive connexins that we have looked at that the Hill coefficient versus CO_2_ is >4. Hemichannels are of course hexameric so there is potential for 6 CO_2_ molecules to be bound and extensive cooperativity. We have modified the text to give greater context.

(d) A further remark to Figure 2 is that it does not contain any experiment showing the effect of Cx43 hemichannel inhibition with a reliable HC inhibitor such as Gap26, which is only used in the penultimate illustration of Figure 10. Gap26 should be used in Figure 2 and most of the other figures to show evidence of HC contribution. The lanthanum ions used in Figure 9 are a very non-specific hemichannel blocker and should be replaced by experiments with Gap26.

We have addressed the first part of this comment above.

We agree that La^3+^ blocks all hemichannels, but in the context of our experiments and the controls we have performed it is entirely adequate and supports our conclusions. Our controls show (mentioned above and below) show that the expression of Cx43 is absolutely required for CO_2_-dependent ATP release (and dye loading). In Figure 9 our use of La^3+^ was to show the presence of a constitutively open Cx43 mutant hemichannel. Gap26 would add little to this. Our further controls show that with expression of Cx43^WT^ La^3+^ did nothing to the ATP signal under baseline conditions (20 mmHg) supporting our conclusion that the mutant channels are constitutively open.

(e) As the experiments of Figure 2 form the basis of what is to follow, the above remarks cast doubt on the robustness of the experiments and the data produced.

We disagree, our results are extremely robust: (1) we have used three independent assays confirm the presence of the response; (2) parental HeLa cells do not release ATP, dye load or show large conductance changes to CO_2_ showing the absolute requirement for expression of Cx43; (3) mutations of Cx43 (in the carbamylation motif) alter the CO_2_ evoked ATP release and dye loading giving further confirmation of Cx43 as the conduit for ATP release and dye loading; and (4) we use standard positive controls (0 Ca^²^, high K) to confirm cells still have functional channels for those mutations that modified CO_2_ sensitivity.

(f) The sentence "Cells transfected with GRAB-ATP only, showed ... " should be

modified to "In contrast, cells not expressing Cx43 showed no responses to any applied CO2 concentration as concluded from GRAB-ATP experiments"

We have modified the text.

(3) Page 5 and Figures 3 & 4:(a) Figure 3 illustrates results obtained with mutations of 4 distinct Lys residues. However, the corresponding legend indicates mutations that are different from the ones shown in the corresponding illustrations, making it impossible to reliably understand and interpret the results shown in panels A-E.

Thanks for pointing this out. Our apologies, we modified the figure so that the order of the images matched the order of the graph (and the legend) but then forgot to put the new version of the figure in the text. We have now corrected this so that Figure and legend match.

(b) Figure 4 lacks control WT traces!

The controls for this (showing that parental HeLa cells do not release ATP in response to CO_2_ or depolarisation) are shown in Figure 2.

(c) Figure 4, Supplement 1: High Hill coefficients of 10 are shown here, but they are not discussed anywhere, as is also the case for the remark on p.4. A Hill steepness of 10 is huge and points to many processes potentially involved. As reported above, these data are floating around in the manuscript without any connection.

Yes, we agree this is very high and surprising. It may reflect as mentioned above the hexameric nature of the channel and that 4 Lys residues seem to be involved. We have used this equation to give some quantitative understanding of the effect of the mutations on CO_2_ sensitivity and still think this is useful. We have no further evidence to interpret these values one way or the other.

(4) Page 6: Carbamate bridges are proposed to be formed between K105 and K144, and between K109 and K234. The first three of these Lysine residues are located in the 55aa long cytoplasmic loop of Cx43, while K234 is in the juxta membrane region involved in tubulin interactions. Both K144 and K234 are involved in Cx43 HC inhibition: K144 is the last aa of the L2 peptide (D119-K144 sequence) that inhibits Cx43 hemichannels while K234 is the first aa of the TM2 peptide that reduces hemichannel presence in the membrane (sequence just after TM4, at the start of the C-tail). This context should be added to increase insight and understanding of the CO2 carbamylation effects on Cx43 hemichannel opening.

Thanks for suggesting this. We have added some discussion of CT to CL interactions in the context of regulation by pH and [Ca^2+^].

(5) Page 7: The Cx43 ODDD A44V and L90V mutations lead to loss of pCO2 sensitivity in dye loading and ATP assays. However, A44V located in EL1 is reportedly associated with Cx43 HC activation, while L90V in TM2 is associated with HC inhibition. Remarkably, these mutations are focused on non-Lys residues, which brings up the question of how to link this to the paper's main thread.

This follows the pattern that we have seen for other mutations such as A40V, A88V in Cx26 and several CMTX mutations of Cx32. Our cryoEM structures of Cx26 suggest that these mutations alter the flexibility of the molecule and hence abolish CO_2_ sensitivity. We have reworded the text to avoid giving the impression that there is a demonstrated link between loss of CO_2_ sensitivity of Cx43 and pathology.

(6) Page 8: HCs constitutively open - 'constutively' perhaps does not have the best connotation as it is not related to HC constitution but CO2 partial pressure.

Yes, we agree and have reworded this.

(7) Page 9: "in all subtypes" -> not clear what is meant - do you mean "in all cell types"?

We agree this is unclear -it refers to all astrocytic subtypes. We have amended the text.

(8) Page 10: Composition of hypocapnic recording solution: bubbling description is incomplete "95%O2/5%" and should be "95%O2/5%CO2".

Changed.

(9) Page 11: Composition of zero Ca^²⁺^ hypocapnic recording solution: perhaps better to call this "nominally Ca^²⁺^-free hypocapnic recording solution" as no Ca^²⁺^ buffer is included in this solution

Thanks for pointing this out. We did in fact add 1 mM EGTA to the solutions but omitted this from the recipe, this has now been corrected.

(10) Page 11: in M&M I found that the NaHCO3- is lowered to 10 mM in the zero Ca^²⁺^condition, while the control experimental condition has 26 mM NaHCO3-. The zero Ca condition should be kept at a physiologically normal 26 mM NaHCO3- concentration, so why was this done? Lowering NaHCO3- during hemichannel stimulation may result in smaller responses and introduce non-linearities.

For the dye loading we used 20 mmHg as the baseline condition and increased PCO_2_ from this. Hence for the zero Ca^2+^ positive control we modified the 20 mmHg hypocapnic solution by substituting Mg^2+^ for Ca^2+^ and adding EGTA. We have modified the text in the Methods to clarify this.

Further remarks on the figures:(1) Figure 2A: Add 20 & 70 mmHg to the images, to improve the readability of this illustration.

Done

(2) Figure 3: WT responses are shown in panel F, but experimental data (images and curves) are lacking and should be included in a revised version.

The wild type data is shown in Fig 2A. We have some sympathy for the comment, but we felt that Fig 2 should document CO_2_ sensitivity, and then the subsequent Figs should analyse its basis. Hence the separation of Cx43^WT^ data from the mutant data. In panel F, we state that we have recalculated the WT data from Fig 2A to allow the comparison.

(3) Figures 4, 6, 8: Color codes for mmHg CO_2_ pressure make reading these figures difficult; perhaps better to add mmHg values directly in relation to the traces.

We have considered this suggestion but feel that the figures would become very cluttered with the additional labelling.

(4) I wouldn't use colored lines when not necessary, e.g., Figure 9 100 µM La3+; Figure 10 (add 20->35 mmHg PCO2 switch; add scrGap26 above blue bars); Figure 11C & D.

We agree and can see that in Figs 9 and 10 this muddles our colour scheme in other figures so have modified these figures. There was not space to put the suggested labels.

(5) The mechanism of increased HC opening is not clear.

We agree and have discussed various options and the analogy with what we know about Cx26. Ultimately new cryo-EM data is required.

(6) Figure 10: 35G/35S are weird abbreviations for 35 mmHg Gap26 and scrambled Gap26.

Yes, but we used these to fit into the available space.

(7) Figure 11, legend: '20 mmHg PCO2 for each transfection for 70 mmHg PCO2'. It is not clear what is meant here.

Thanks for pointing this out, we have reworded this to ensure clarity.